# An L-2-hydroxyglutarate biosensor based on specific transcriptional regulator LhgR

Zhaoqi Kang[1], Manman Zhang[2], Kaiyu Gao[1], Wen Zhang[3], Wensi Meng[1], Yidong Liu[1], Dan Xiao[1], Shiting Guo[1], Cuiqing Ma[1], Chao Gao [1✉] & Ping Xu [4✉]

L-2-Hydroxyglutarate (L-2-HG) plays important roles in diverse physiological processes, such as carbon starvation response, tumorigenesis, and hypoxic adaptation. Despite its importance and intensively studied metabolism, regulation of L-2-HG metabolism remains poorly understood and none of regulator specifically responded to L-2-HG has been identified. Based on bacterial genomic neighborhood analysis of the gene encoding L-2-HG oxidase (LhgO), LhgR, which represses the transcription of *lhgO* in *Pseudomonas putida* W619, is identified in this study. LhgR is demonstrated to recognize L-2-HG as its specific effector molecule, and this allosteric transcription factor is then used as a biorecognition element to construct an L-2-HG-sensing FRET sensor. The L-2-HG sensor is able to conveniently monitor the concentrations of L-2-HG in various biological samples. In addition to bacterial L-2-HG generation during carbon starvation, biological function of the L-2-HG dehydrogenase and hypoxia induced L-2-HG accumulation are also revealed by using the L-2-HG sensor in human cells.

[1] State Key Laboratory of Microbial Technology, Shandong University, Qingdao, People's Republic of China. [2] Tianjin Key Laboratory of Radiation Medicine and Molecular Nuclear Medicine, Department of Radiobiology, Institute of Radiation Medicine of Chinese Academy of Medical Science and Peking Union Medical College, Tianjin, People's Republic of China. [3] Center for Gene and Immunotherapy, The Second Hospital of Shandong University, Jinan, People's Republic of China. [4] State Key Laboratory of Microbial Metabolism, Joint International Research Laboratory of Metabolic and Developmental Sciences, and School of Life Sciences and Biotechnology, Shanghai Jiao Tong University, Shanghai, People's Republic of China. ✉email: jieerbu@sdu.edu.cn; pingxu@sjtu.edu.cn

L-2-Hydroxyglutarate (L-2-HG) is an important metabolite in various domains of life. In mammals and plants, it is produced by lactate dehydrogenase (LDH) and malate dehydrogenase (MDH)-mediated 2-ketoglutarate (2-KG) reduction under hypoxic conditions[1–5]. In microorganisms, it is a metabolic intermediate of glutarate catabolism produced by a glutarate hydroxylase, CsiD[6–8]. L-2-HG dehydrogenase (L2HGDH) or L-2-HG oxidase (LhgO), an FAD-containing oxidoreductase that converts L-2-HG to 2-KG, plays an indispensable role in the catabolism of L-2-HG[7,9]. While extensive efforts have been devoted to investigating L-2-HG anabolism and catabolism, the molecular machinery that specifically senses L-2-HG and regulates its metabolism has not been identified until now.

L-2-HG is an inhibitor of 2-KG dependent dioxygenases with specific pro-oncogenic capabilities[10,11]. Thus, this oncometabolite is viewed as a biomarker for a variety of cancers and its rapid and sensitive measurement in body fluids is of clinical significance[12–15]. Importantly, L-2-HG also has endogenous functions in healthy animal cells. For example, this compound was recently identified to aid the proliferation and anti-tumorigenic abilities of CD8[+] T-lymphocytes[16], to contribute to relieving the cellular reductive stress[3], and to coordinate glycolytic flux with epigenetic modifications[17]. Considering the various roles of L-2-HG in cell metabolism, the development and optimization of real-time monitoring assays for this metabolite in living cells are required.

Liquid chromatography-tandem mass spectrometry (LC-MS/MS)[18,19] and gas chromatography-tandem mass spectrometry (GC-MS/MS)[20,21] are often used to assess the extracellular concentrations of L-2-HG. These state-of-the-art methods are time-consuming, expensive to perform, and require highly skilled personnel. In addition, these destructive methods are also incompatible with real-time monitoring of the fluctuations of L-2-HG concentrations in intact living cells. In this study, we identify and characterize LhgR, an L-2-HG catabolism regulator in *Pseudomonas putida* W619. Mechanistically, LhgR represses the transcription of LhgO encoding gene *lhgO*. L-2-HG is a specific effector molecule of LhgR and prevents LhgR binding to the promoter region of *lhgO*. We then report the development and application of the LhgR-based L-2-HG biosensor via Förster resonance energy transfer (FRET), a technology widely applied to investigate temporal dynamics of various small molecules, such as potassium[22,23], glycine[24], and cAMP[25,26]. As-designed sensor quantitatively responds to L-2-HG concentrations in various biological samples with high accuracy and precision. We also use this biosensor to identify the carbon starvation-induced L-2-HG production in bacteria and to demonstrate hypoxia-induced L-2-HG production by LDH and MDH in human cells. Therefore, the LhgR-based biosensor can prove to be a useful tool for real-time measurement of the L-2-HG concentrations in living cells.

## Results

**LhgR regulates L-2-HG catabolism**. In this study, bacteria containing LhgO encoding gene *lhgO* were selected to study the regulation of L-2-HG metabolism. Homologs of LhgO can be found in 612 different bacterial strains. Similarly organized chromosomal clusters are found in many bacterial genomes, which contain various combinations of genes related to glutarate metabolism (*csiD*, *lhgO*, *gabT*, *gabD*, and *gabP*) (Fig. 1a). In *Pseudomonas putida* KT2440, the glutarate regulon is regulated by allosteric transcription factor CsiR, which is encoded upstream of *csiD*[27]. The glutarate sensing allosteric transcription factor CsiR and its cognate promoter were cloned into broad host range vectors to create a glutarate biosensor[28]. Interestingly, a different pattern of *lhgO* gene neighborhood was observed in a few species that do not contain *csiD* homologs (Fig. 1a). For example, a gene

encoding a GntR family protein, *lhgR*, was found directly upstream of *lhgO* in *P. putida* W619. The absence of *csiD* gene related to glutarate catabolism made us to reason that *lhgO* of *P. putida* W619 might be solely involved in L-2-HG metabolism and be L-2-HG inducible.

The *lhgO* gene in *P. putida* W619 was cloned into pME6032 vector and transferred into *P. putida* KT2440 (Δ*lhgO*). As shown in Fig. 1b, c, the complement of *lhgO* in *P. putida* W619 could restore glutarate and L-2-HG utilization abilities of *P. putida* KT2440 (Δ*lhgO*), confirming that *lhgO* encodes a functional L-2-HG catabolic enzyme. To identify the function of LhgR in *P. putida* W619, the gene segment F2-*lhgR*-F1-*lhgO*, which contains the promoter of *lhgR* (F2), *lhgR*, the promoter of *lhgO* (F1), and *lhgO*, was cloned into pME6032 vector, and the resulting plasmid was transferred into different derivatives of *P. putida* KT2440 (Fig. 1d). As shown in Fig. 1e, exogenous L-2-HG, but not its mirror-image enantiomer D-2-HG, can induce the expression of *lhgO* in the gene segment F2-*lhgR*-F1-*lhgO* and restore LhgO activity in *P. putida* KT2440 (Δ*csiR*Δ*lhgO*). In addition, the activity of LhgO was also detected in *P. putida* KT2440 (Δ*csiR*Δ*lhgO*) harboring pME6032-F2-*lhgR*-F1-*lhgO* when cultured with glutarate as the sole carbon source. However, no activity of LhgO was detected in *P. putida* KT2440 (Δ*csiR*Δ*csiD*Δ*lhgO*), in which the key gene responsible for L-2-HG production from glutarate was deleted. These results indicated that LhgR represses the expression of LhgO and L-2-HG, but neither D-2-HG nor glutarate can serve as the effector molecule of LhgR.

**LhgR specifically responds to L-2-HG**. To determine whether LhgR directly interacts with the promoter region of *lhgO*, LhgR in *P. putida* W619 was overexpressed in *E. coli* BL21(DE3) and purified by Ni-chelating chromatography (Fig. 2a). Based on the results of gel filtration and sodium dodecyl sulfate-polyacrylamide gel electrophoresis (SDS-PAGE), LhgR behaved as a dimer (Fig. 2b). Subsequently, electrophoretic mobility shift assays (EMSAs) were conducted using *lhgO* promoter (F1) and purified LhgR. As shown in Fig. 2c, LhgR bound to F1 in a concentration-dependent manner. LhgR completely shifted fragment F1 gel band when an 8-fold molar excess was used. A DNase I footprinting assay was also performed using purified LhgR and fragment F1. A protected region containing palindromic $N_yGTN_xACN_y$ consensus binding motif of GntR-family allosteric transcription factor[29], 5′-TA**GT**CTG**AC**AA-3′, was observed (Fig. 2d). In addition, LhgR also bound to its promoter (F2) in EMSAs. A similar consensus binding motif, 5′-TT**GT**CTG**AC**AA-3′, was protected in DNase I footprinting assay (Supplementary Fig. 1a-b).

Effects of L-2-HG, D-2-HG, glutarate, 2-KG, L-lysine, 5-aminovalerate, and succinate on LhgR binding to the *lhgO* promoter region F1 were also assessed by EMSAs. The release of LhgR from fragments F1 was observed only in the presence of L-2-HG (Fig. 2e). These results suggested that L-2-HG can specifically prevent the binding of LhgR to the promoter of *lhgO* and induce its expression. LhgR may help *P. putida* W619 to specifically sense L-2-HG generated by intracellular metabolism or present in habitats and stimulate the catabolism of L-2-HG. In addition, LhgR may self-repress its expression and L-2-HG can also contribute to inducing the expression of *lhgR* (Supplementary Fig. 1c).

**Design and optimization of the L-2-HG-sensing reporter**. FRET sensors, which combine a ligand-binding moiety and a pair of donor-acceptor fluorescent pair, allow measurement of ligand concentrations based on the ligand-binding induced changes of

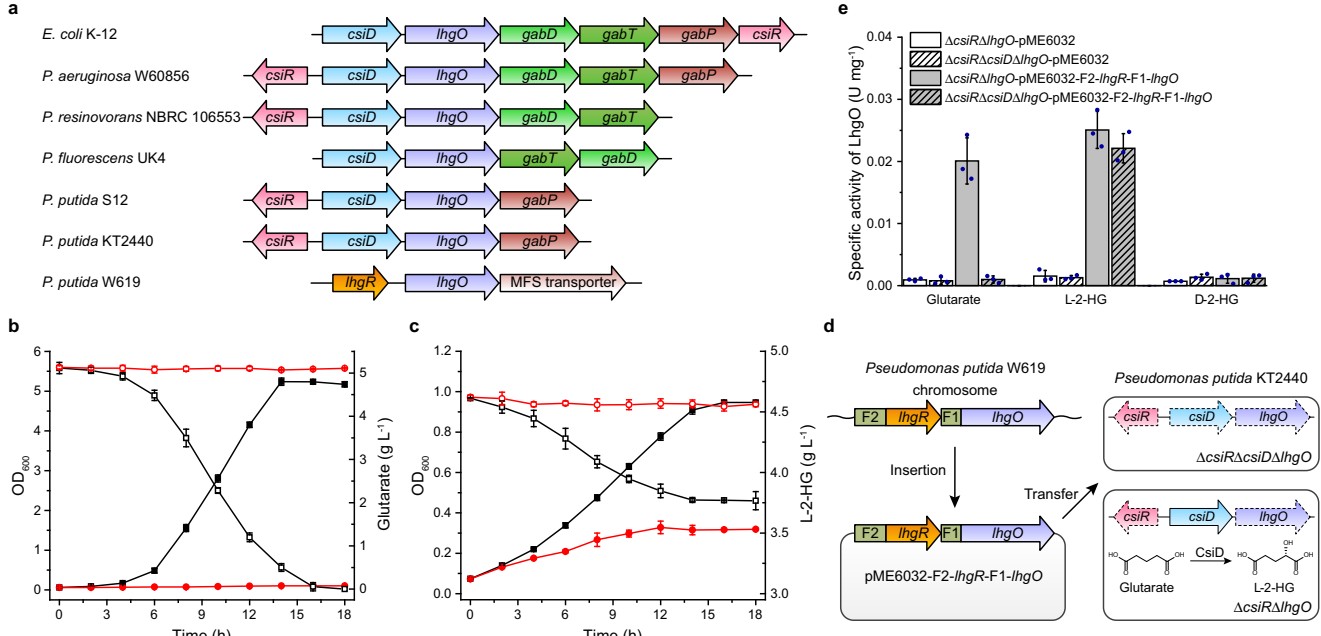

**Fig. 1 Regulation of L-2-HG catabolism by LhgR in *P. putida* W619. a** Schematic representation of genomic neighborhood analysis of *lhgO* in different bacteria. Orthologs are shown in the same color and the direction of gene transcription is indicated by arrows. CsiR, GntR family allosteric transcription factor regulating glutarate catabolism; CsiD, glutarate hydroxylase; LhgO, L-2-HG oxidase; GabD, succinate semialdehyde dehydrogenase; GabT, 4-aminobutyrate aminotransferase; GabP, 4-aminobutyrate transporter. **b, c** Growth of the derivatives of *P. putida* KT2440 in MSMs with glutarate (**b**) or L-2-HG (**c**) as the sole carbon source. Growth (closed symbols) and the consumption of carbon source (open symbols) of *P. putida* KT2440 (Δ*lhgO*) harboring plasmid pME6032-*lhgO* (black lines with squares) and *P. putida* KT2440 (Δ*lhgO*) harboring empty plasmid pME6032 (red lines with circles) were measured in MSMs containing 5 g L⁻¹ glutarate (**b**) or L-2-HG (**c**) as the sole carbon source. **d** Schematic representation of the construction of pME6032-F2-*lhgR*-F1-*lhgO* and its transfer into *P. putida* KT2440 (Δ*csiR*Δ*lhgO*) and *P. putida* KT2440 (Δ*csiR*Δ*csiD*Δ*lhgO*) by electroporation. The deleted genes in *P. putida* KT2440 are indicated by dashed lines. The reaction catalyzed by CsiD is also demonstrated. **e** The activities of LhgO in *P. putida* KT2440 (Δ*csiR*Δ*lhgO*) and *P. putida* KT2440 (Δ*csiR*Δ*csiD*Δ*lhgO*) harboring either plasmid pME6032-F2-*lhgR*-F1-*lhgO* or empty plasmid pME6032 grown in MSM with glutarate, L-2-HG, or D-2-HG as the sole carbon source. All data shown are means ± standard deviations (s.d.) (*n* = 3 independent experiments).

FRET efficiency[22–26]. In this study, the L-2-HG-sensing fluorescent reporter (LHGFR) was constructed by fusion of the optimized cyan and yellow fluorescent protein variants, mTFP[30] and Venus[31], to the N-terminus and C-terminus of LhgR (Supplementary Fig. 2). This first LHGFR was named LHGFR$_{0N0C}$, where the subscript indicates the number of amino acids truncated from the N-terminus and C-terminus of LhgR. Subsequently, LHGFR$_{0N0C}$ was overexpressed in *E. coli* BL21(DE3) and purified by a Ni-chelating chromatographic column (Supplementary Fig. 3). Spectra properties of LHGFR$_{0N0C}$ reveal the addition of L-2-HG could reduce the emission peak at 492 nm of mTFP and increase the emission peak at 526 nm of Venus (Supplementary Fig. 4). Thus, the conformational change of LhgR after the L-2-HG binding may lead to a shortened relative distance and/or favorable orientation of mTFP and Venus, resulting in an increase in FRET (Fig. 3a). In addition, L-2-HG increased the emission ratio of Venus to mTFP in a dose-dependent manner, with a maximum ratio change ($\Delta R_{max}$) of 11.47 ± 0.38%, an apparent dissociation constant ($K_d$) of 2.74 ± 0.73 μM, and a Hill slope close to 1 (Fig. 3b).

To increase the magnitude of responses, LHGFR was optimized by truncating N-terminal and C-terminal amino acids of LhgR or by adding a series of artificial linkers between LhgR and various fluorescent proteins[23,32–34] (Fig. 3c and Supplementary Fig. 5). Truncation of three to seven C-terminal amino acids in LhgR could significantly increase $\Delta R_{max}$ of the sensor (Fig. 3c, d and Supplementary Fig. 5). Among the five sensors with increased response magnitude values, LHGFR$_{0N7C}$ exhibited the highest $\Delta R_{max}$ of 60.37 ± 1.30% and $K_d$ of 7.22 ± 0.38 μM (Fig. 3d and

Supplementary Fig. 6a–e). In addition, LHGFR$_{0N3C}$ was also a promising sensor with a high $\Delta R_{max}$ of 56.13 ± 0.29% and a high $K_d$ of 29.33 ± 1.24 μM (Fig. 3d and Supplementary Fig. 6a).

Then, the properties of LHGFR$_{0N3C}$ and LHGFR$_{0N7C}$ were also investigated. Both LHGFR$_{0N3C}$ and LHGFR$_{0N7C}$ behave as tetramers and have lost the ability to bind DNA (Supplementary Fig. 7). L-2-HG binding increased FRET between the fluorophores in LHGFR$_{0N3C}$ and LHGFR$_{0N7C}$ (Supplementary Fig. 8). D-Lactate, L-lactate, as well as a set of intermediates of TCA cycle and L-lysine catabolism, were used to examine the specificity of LHGFR$_{0N3C}$ and LHGFR$_{0N7C}$. None of D-lactate, L-lactate, oxaloacetate, citrate, isocitrate, 2-KG, succinate, fumarate, cis-aconitate, L-malate, pyruvate, L-lysine, 5-aminovalerate, and glutarate induced the emission ratio changes of LHGFR$_{0N3C}$ or LHGFR$_{0N7C}$ (Fig. 3e, f). The addition of these compounds also had no influence on the response of LHGFR$_{0N3C}$ or LHGFR$_{0N7C}$ to L-2-HG (Supplementary Fig. 9a–b). Both LHGFR$_{0N3C}$ and LHGFR$_{0N7C}$ exhibited much higher affinity for L-2-HG than that for D-2-HG (Supplementary Fig. 9c–d and Supplementary Table 1). The limits of detection (LODs) of LHGFR$_{0N3C}$ and LHGFR$_{0N7C}$ for L-2-HG and D-2-HG were 4.34 μM and 872.59 μM, 0.70 μM and 128.34 μM, respectively (Supplementary Table 1). The dose-response curves of LHGFR$_{0N3C}$ and LHGFR$_{0N7C}$ for L-2-HG in the absence or presence of D-2-HG, 2-KG, and ATP were also assayed (Supplementary Fig. 9e–j). Similar LODs, $K_d$ values, and $\Delta R_{max}$ of LHGFR$_{0N3C}$ were detected (Supplementary Fig. 9e–j and Supplementary Table 1). Kinetic analyses of LHGFR$_{0N3C}$ and LHGFR$_{0N7C}$ were performed and values for the association rate constant ($k_{on}$) and dissociation rate

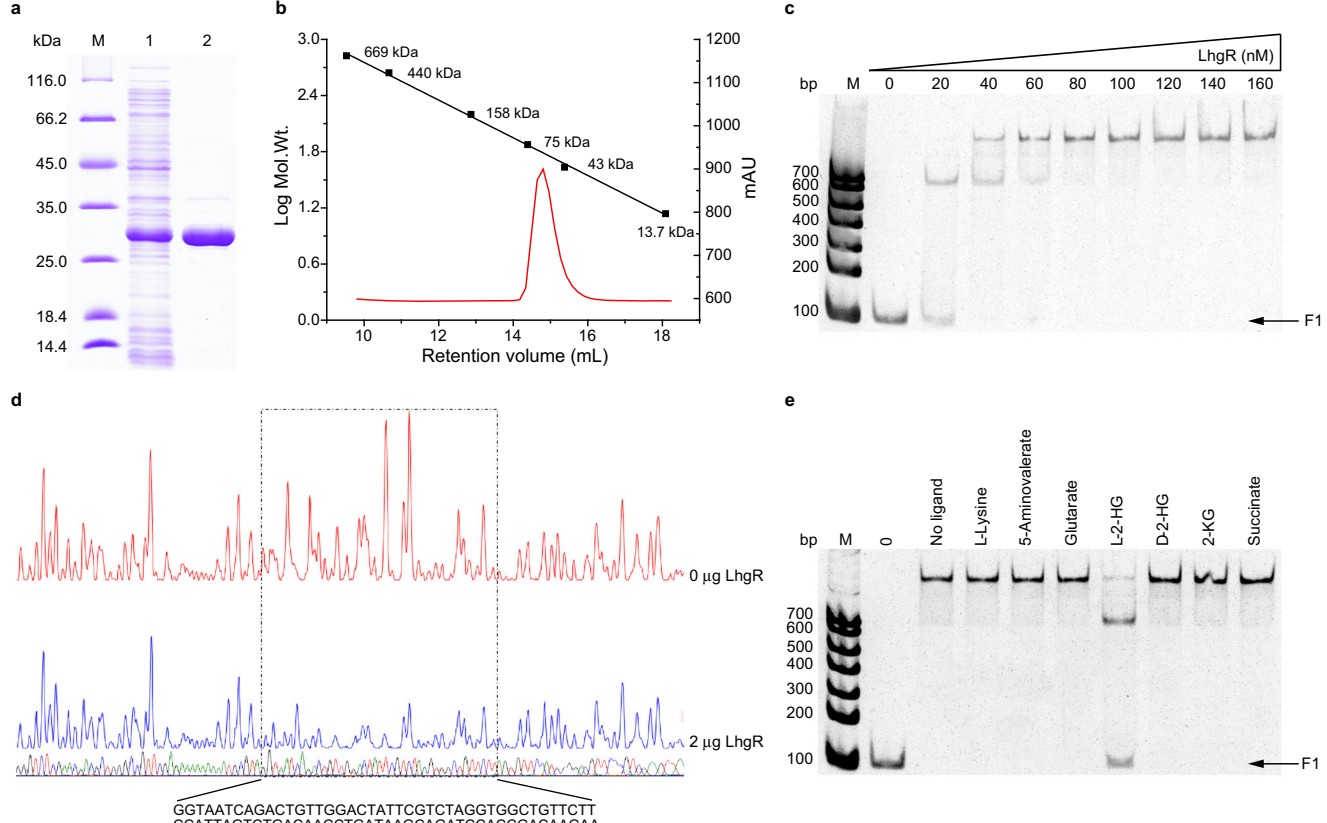

**Fig. 2 Purification and characterization of LhgR. a** SDS-PAGE analysis of the purification of LhgR. Lane M, molecular weight markers; lane 1, crude extract of *E. coli* BL21(DE3) harboring pETDuet-*lhgR*; lane 2, purified His₆-tagged LhgR using a HisTrap column. **b** Gel-filtration chromatography of the purified LhgR with the Superdex 200 10/300 GL column. Red curve, the chromatogram of purified LhgR; Black line, a standard curve for protein molecular mass standards. **c** LhgR can bind to the *lhgO* promoter region. F1 fragment containing the *lhgO* promoter region (10 nM) was titrated by purified LhgR (0, 20, 40, 60, 80, 100, 120, 140, 160 nM). Lane M, molecular weight markers. **d** DNase I footprinting analysis of LhgR binding to the *lhgO* promoter region. The F1 fragment was labeled with 6-carboxyfluorescein (FAM) and incubated with 2 μg LhgR (blue line) or without LhgR (red line). The region protected by LhgR is indicated with a dotted box. **e** L-2-HG prevents LhgR binding to the *lhgO* promoter region. EMSAs were carried out with F1 fragment (10 nM) and purified LhgR (60 nM) in the absence of any other tested compounds (No ligand) and in the presence of 50 mM different compounds. Lane M was the molecular weight marker; lane 0 without LhgR was used as the control.

constant ($k_{off}$) of LHGFR$_{0N3C}$ and LHGFR$_{0N7C}$ were determined to be $5.50 \times 10^{-1}$ μM$^{-1}$ s$^{-1}$ and 15.75 s$^{-1}$, $2.84 \times 10^{-1}$ μM$^{-1}$ s$^{-1}$ and 5.79 s$^{-1}$, respectively (Supplementary Fig. 10a–d). Effects of temperature on LHGFR$_{0N3C}$ and LHGFR$_{0N7C}$ were analyzed, respectively, and the affinity of LHGFR$_{0N3C}$ to L-2-HG remained unaffected from 25 °C to 37 °C (Supplementary Fig. 10e–h). L-2-HG-dependent emission ratio changes of LHGFR$_{0N3C}$ or LHGFR$_{0N7C}$ were reversible by L-2-HG oxidation catalyzed by 5 μM LhgO (Fig. 3g, h and Supplementary Fig. 10i-j) and both biosensors were stable for the detection of L-2-HG from pH 6.0 to 8.0 (Supplementary Fig. 10k-l).

**Characterization of LHGFR in biological samples.** Next, we investigated whether LHGFR could be used to quantify L-2-HG concentrations in different biological samples. When L-2-HG with increasing concentrations (0 to 2 mM) were added into serum and urine samples of healthy adults, the response curves were nearly identical with that in assay buffer for both LHGFR$_{0N3C}$ and LHGFR$_{0N7C}$ (Fig. 4a–d, Supplementary Fig. 6a and Fig. 6e). Thus, quantitative determination of L-2-HG could be conducted by mixing the target samples with LHGFR and measuring the emission ratios with a conventional fluorescence microplate reader. Based on the response curves established for L-2-HG quantification, both biosensors were used to assay the concentrations of L-2-HG in human serum and urine (Fig. 4e, f).

The results of LHGFR$_{0N3C}$ and LHGFR$_{0N7C}$ showed close agreement with the results of LC-MS/MS, the current standard method for clinical assays of L-2-HG (Fig. 4e, f and Supplementary Table 2).

In a previous report, L-2-HG was identified to be a metabolic intermediate of glutarate metabolism in *P. putida* KT2440[7]. LHGFR$_{0N3C}$ and LHGFR$_{0N7C}$ also exhibited high accuracy in the quantification of L-2-HG in bacterial culture medium (Fig. 4g-i). When cultured in the medium containing 20 mM glucose and 5 mM glutarate as carbon sources, the growth of *P. putida* KT2440 (Δ*lhgO*) was significantly delayed, which might be due to the possible toxicity of accumulated L-2-HG (Fig. 4j). Nearly identical results of L-2-HG quantification were also obtained by either using LHGFR$_{0N3C}$, LHGFR$_{0N7C}$, or using LC-MS/MS (Fig. 4k, l and Supplementary Fig. 11). Mutual corroboration between the two biosensors further confirmed their applicability in in vitro L-2-HG quantification of various biological samples. In addition, the response curves of LHGFR$_{0N3C}$ and LHGFR$_{0N7C}$ for D-2-HG in serum, urine, and bacterial culture medium were also determined (Supplementary Fig. 12). The LODs of LHGFR$_{0N3C}$ and LHGFR$_{0N7C}$ for D-2-HG were more than 100-fold higher than those for L-2-HG (Supplementary Table 3 and Table 4).

**Monitoring L-2-HG fluctuations in living bacteria by LHGFR.** We also investigated whether the LhgR-based L-2-HG sensor

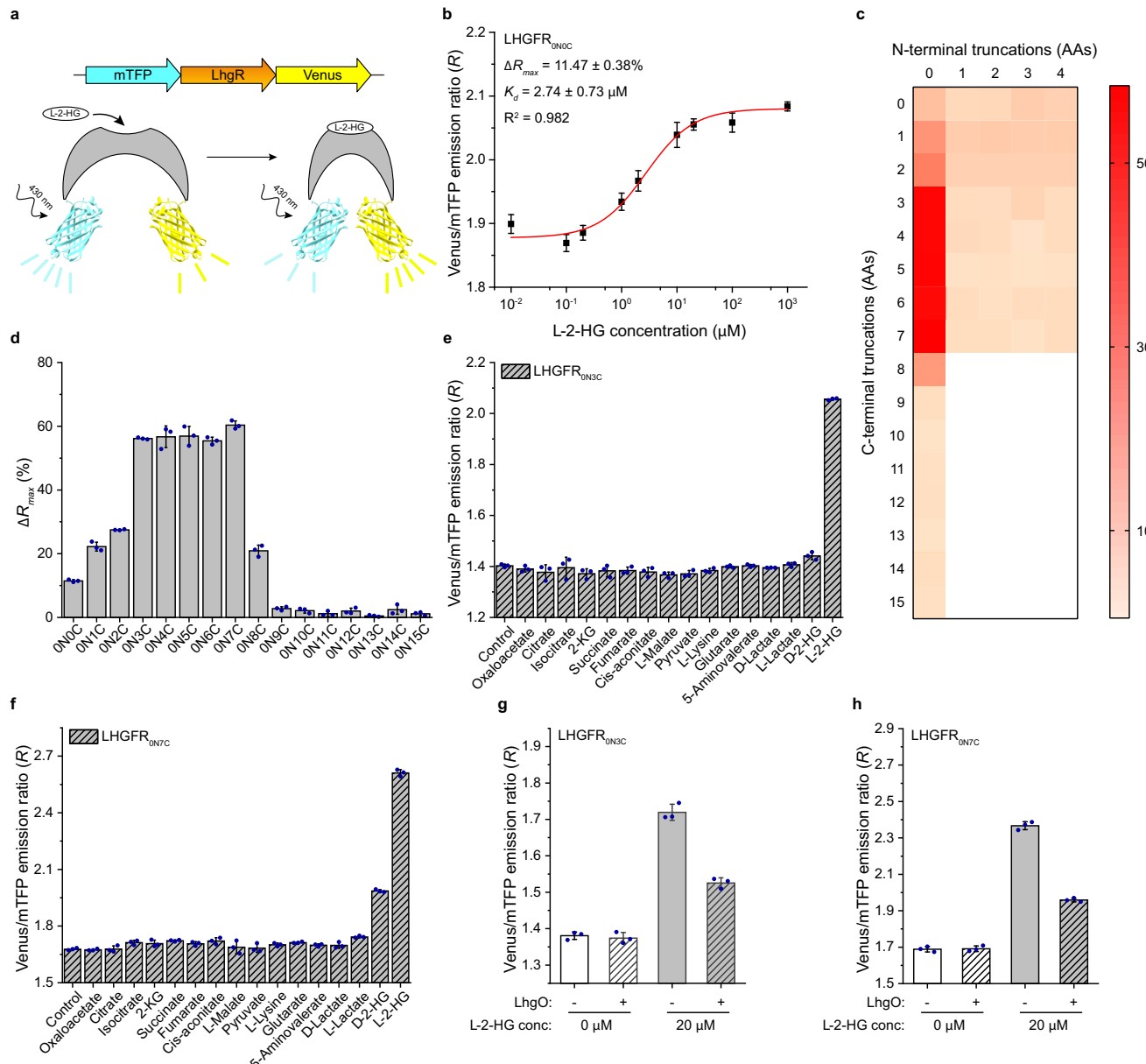

**Fig. 3 Design and optimization of the LHGFR. a** Schematic representation of the predicted conformational change of LhgR-based L-2-HG biosensor LHGFR in the absence or presence of L-2-HG. In particular, schematic representations of the tertiary structure of Venus and mTFP shown were predicted based on the respective protein sequences. **b** Dose-response curve of purified LHGFR$_{0N0C}$ for increasing concentrations (10 nM to 1 mM) of L-2-HG in 50 mM Tris-HCl buffer (pH 7.4). The emission ratio of Venus to mTFP increased (430 nm excitation) after L-2-HG binding. **c** Heap map of the truncating the N-terminal and C-terminal amino acids of LhgR to $\Delta R_{max}$. The color indicates the value of $\Delta R_{max}$ and white indicates the untested variants. **d** Comparison of the $\Delta R_{max}$ of a set of L-2-HG biosensor variants based on the C-terminal amino acid truncated of LhgR. **e, f** Specificities of the purified LHGFR$_{0N3C}$ (**e**) and LHGFR$_{0N7C}$ (**f**). The emission ratio changes of both biosensors were measured in the presence of 240 μM D-lactate, L-lactate, L-2-HG, D-2-HG, or different intermediates of the TCA cycle and L-lysine catabolism. **g, h** Reversal of L-2-HG binding with LHGFR by conversion of L-2-HG to 2-KG. The emission ratio of purified LHGFR$_{0N3C}$ (**g**) and LHGFR$_{0N7C}$ (**h**) was recorded in the absence and presence of 20 μM L-2-HG before and after the addition of 5 μM purified LhgO for 25 min. All data shown are means ± s.d. ($n = 3$ independent experiments).

LHGFR could detect possible variations of L-2-HG in living bacteria. LHGFR$_{0N3C}$ and LHGFR$_{3N7C}$, a control biosensor that did not respond to L-2-HG in vitro (Fig. 3c), were expressed in E. coli BL21 (DE3). Exogenous L-2-HG was added to the culture system of E. coli BL21(DE3) to achieve final concentrations between 0 and 10 mM and the emission ratio was continuously recorded. As shown in Fig. 5a, exogenous L-2-HG could increase emission ratios of LHGFR$_{0N3C}$ in a dose-dependent manner. The apparent $K_d$ of LHGFR$_{0N3C}$ expressed in E. coli BL21(DE3) was determined to be 891.72 ± 32.10 μM by fitting emission ratios against exogenous L-2-

HG concentrations (Fig. 5b). Maturation time lag and degradation of biosensors may be reasons responsible for higher apparent $K_d$ of LHGFR$_{0N3C}$ when expressed in E. coli[35]. The specificity of LHGFR$_{0N3C}$ expressed in E. coli BL21(DE3) was also characterized. As shown in Fig. 5c, only exogenous L-2-HG could significantly increase the emission ratio in E. coli BL21(DE3), while glutarate, D-2-HG, and glucose could not.

Besides being a metabolic intermediate of exogenous glutarate catabolism in P. putida KT2440, L-2-HG is also reported as a metabolite produced from endogenous L-lysine during carbon

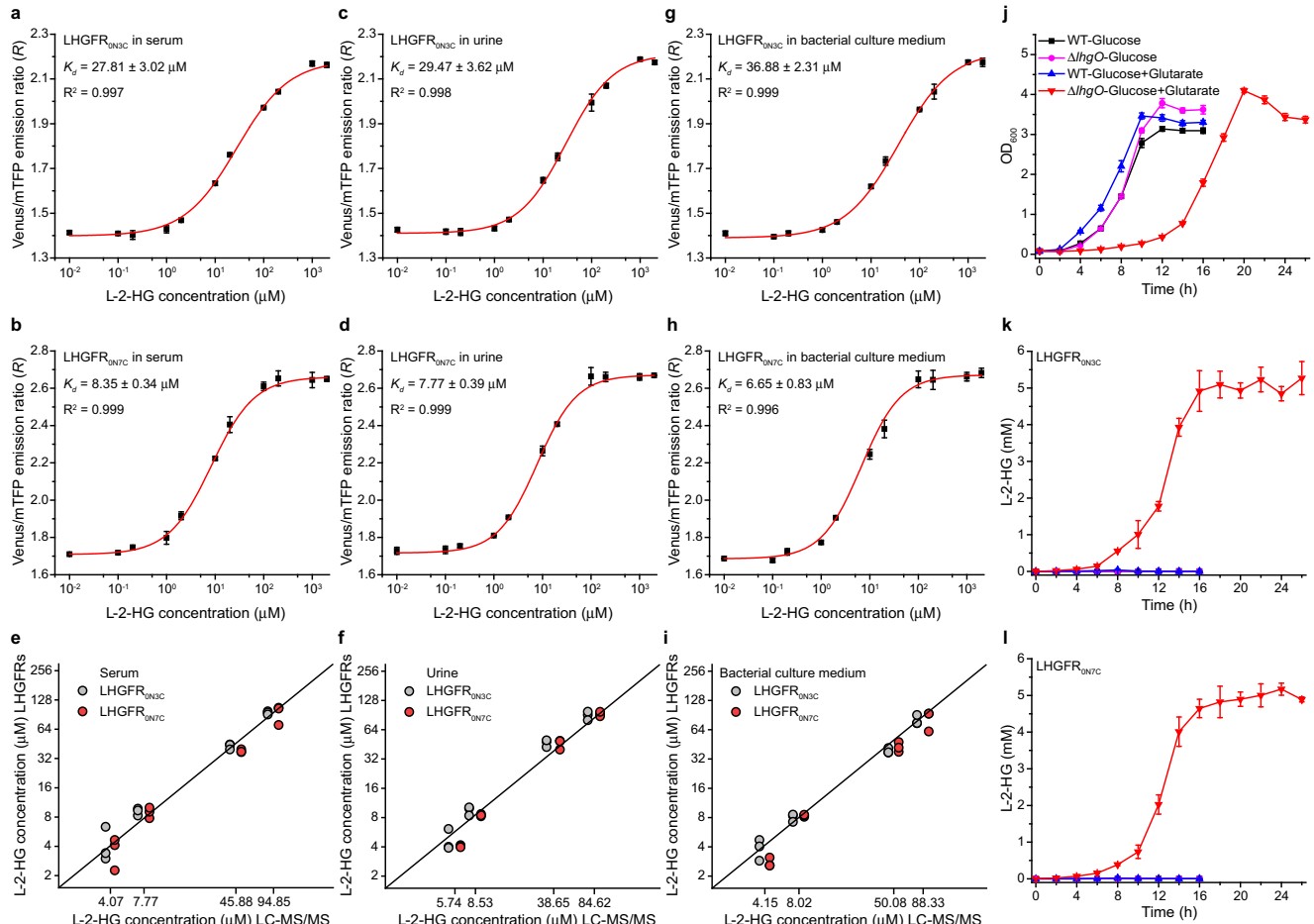

**Fig. 4 Validation of purified LHGFR for determination L-2-HG levels in body fluids and bacterial culture system. a–d** Dose-response curves of purified LHGFR$_{0N3C}$ and LHGFR$_{0N7C}$ for increasing concentrations (10 nM to 2 mM) of L-2-HG in serum (**a, b**) and urine (**c, d**). **e, f** Comparison between the quantitative results of L-2-HG added in serum (**e**) and urine (**f**) by LC-MS/MS and LHGFR. The gray circles and red circles represent the quantitative results of LHGFR$_{0N3C}$ and LHGFR$_{0N7C}$, respectively. Black line indicates a reference line. **g, h** Dose-response curves of purified LHGFR$_{0N3C}$ (**g**) and LHGFR$_{0N7C}$ (**h**) for increasing concentrations (10 nM to 2 mM) of L-2-HG in bacterial culture medium. **i** Comparison between the quantitative results of L-2-HG in bacterial culture medium by LC-MS/MS and LHGFR. **j** Growth of *P. putida* KT2440 and its *lhgO* mutant in medium containing 20 mM glucose and 5 mM glutarate as the carbon sources. **k, l** Determination of extracellular L-2-HG accumulation of *P. putida* KT2440 and its *lhgO* mutant by purified LHGFR$_{0N3C}$ (**k**) and LHGFR$_{0N7C}$ (**l**). All data shown are means ± s.d. (*n* = 3 independent experiments).

starvation of *E. coli*[36,37]. Thus, whether carbon starvation could induce intracellular L-2-HG accumulation of *E. coli* was investigated. As shown in Fig. 5d, no change in the emission ratio was detected when 20 mM glucose was added to the culture system. However, the emission ratio increased during carbon starvation of *E. coli* BL21(DE3), suggesting that carbon starvation induced L-2-HG production. The emission ratios also increased after culturing *E. coli* cells for 6 h with glucose addition (Fig. 5e), which might be due to carbon starvation induced by depletion of exogenous glucose. In addition, the emission ratio increased at the beginning of carbon starvation, reaching a maximum value at 3 h and then decreased to initial levels at 8 h (Fig. 5e). These results confirmed that L-2-HG is a temporary metabolite during carbon starvation and LHGFR$_{0N3C}$ can be used to monitor the real-time change in intracellular L-2-HG concentrations.

To identify whether carbon starvation-induced endogenous L-2-HG production also results from the glutarate hydroxylase activity of CsiD, gene *csiD* was disrupted and LHGFR$_{0N3C}$ was expressed in *E. coli* MG1655(DE3). Gene *lhgO* in *E. coli* MG1655 (DE3) was also disrupted to investigate its role in endogenous L-2-HG catabolism. As expected, the emission ratio of

LHGFR$_{0N3C}$ in *E. coli* MG1655(DE3) (Δ*csiD*) remained unaffected during carbon starvation, whereas disruption of *lhgO* significantly increased the emission ratio of LHGFR$_{0N3C}$ in *E. coli* MG1655(DE3) (Δ*lhgO*) (Fig. 5f), indicating the roles of CsiD and LhgO in endogenous L-2-HG metabolism during carbon starvation. The performance of LHGFR$_{0N7C}$ in monitoring L-2-HG fluctuations in living bacteria was also studied and similar results were acquired (Supplementary Fig. 13). As for the control biosensor LHGFR$_{3N7C}$, no change of emission ratio could be detected in living bacteria for any of the above-mentioned experiments (Supplementary Fig. 14).

**Monitoring L-2-HG production in human cells by LHGFR.** Next, LHGFR$_{0N3C}$, LHGFR$_{0N7C}$, and LHGFR$_{3N7C}$ were expressed in the cytosol of HEK293FT cells. The addition of 10 mM L-2-HG affected mTFP and Venus fluorescence intensities, which caused a non-uniform increase in the emission ratio of LHGFR$_{0N3C}$-expressing single cell (Fig. 6a and Supplementary Movie). The average emission ratio reached a maximum value at 5 min and remained constant during the subsequent confocal imaging period (Fig. 6b). The apparent $K_d$ of LHGFR$_{0N3C}$ expressed in

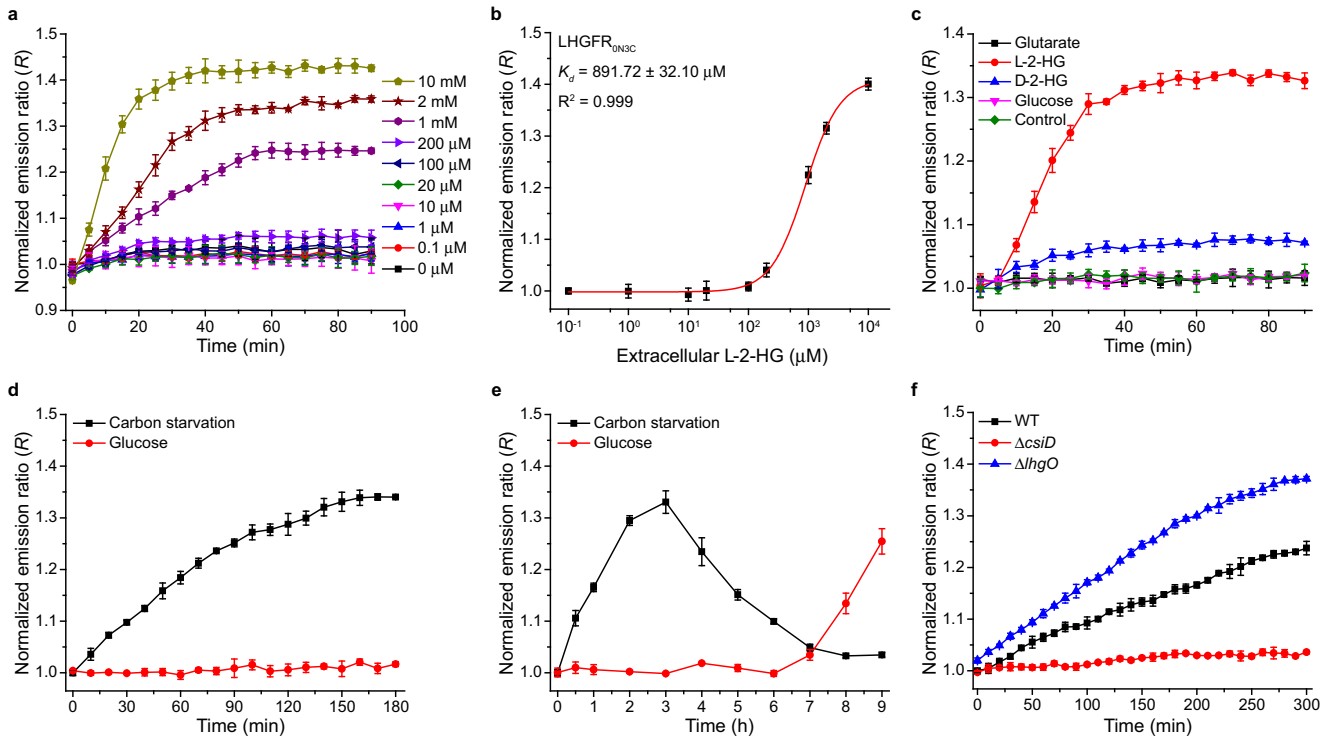

**Fig. 5 Monitoring L-2-HG fluctuations in living bacteria by LHGFR_ON3C. a** Time course of the emission ratio changes of LHGFR_ON3C expressed in *E. coli* BL21(DE3) in response to exogenous L-2-HG addition. All ratios were normalized to the control (ratio in the absence of L-2-HG at time point zero). **b** Normalized dose-response curve of LHGFR_ON3C expressed in *E. coli* BL21(DE3) for increasing concentrations (100 nM to 10 mM) of L-2-HG at time point 60 min. **c** Time course of the emission ratio changes of LHGFR_ON3C expressed in *E. coli* BL21(DE3) in response to the addition of 1 mM glutarate, L-2-HG, D-2-HG, or glucose. All data were normalized to the control (ratio in the absence of any tested compounds at time point zero). **d** Detection of carbon starvation-induced L-2-HG accumulation over time by LHGFR_ON3C expressed in *E. coli* BL21(DE3). Emission ratio changes of LHGFR_ON3C were measured when cultured in carbon starvation medium (black line) and medium with 20 mM glucose (red line). All data were normalized to samples under carbon starvation conditions at time point zero. **e** Long-term detection of L-2-HG fluctuations by LHGFR_ON3C expressed in *E. coli* BL21(DE3). All data were normalized to samples under carbon starvation conditions at time point zero. **f** Identification of the roles of CsiD and LhgO in endogenous L-2-HG metabolism during carbon starvation by LHGFR_ON3C. Emission ratio changes of LHGFR_ON3C expressed in *E. coli* MG1655(DE3) wild-type (black line), *E. coli* MG1655(DE3) (Δ*csiD*) (red line), and *E. coli* MG1655(DE3) (Δ*lhgO*) (blue line) were measured in carbon starvation medium. All data were normalized to time point zero of the wild-type strain. Inconsistent initial emission ratios were detected in bacterial cells under different conditions. The normalized emission ratios were thus used to monitor the changes of L-2-HG in different bacterial cells. All data shown are means ± s.d. (*n* = 3 independent experiments).

HEK293FT cells for L-2-HG was determined to be 43.79 ± 3.05 μM (Fig. 6c). Based on the emission ratio of non-permeabilized HEK293FT cells under physiological conditions, the basal L-2-HG concentration in LHGFR_ON3C-expressing cells was 22.95 ± 11.22 μM (Fig. 6c). Only exogenous L-2-HG could significantly increase the emission ratio of LHGFR_ON3C in 10 μM digitonin-permeabilized HEK293FT cells, suggesting the specificity of the biosensor inside living human cells (Fig. 6d). L2HGDH, the only reported enzyme that is able to catabolize L-2-HG in human cells, is localized in mitochondria[13]. The mitochondrial targeting sequence was appended to LHGFR_ON3C to localize the biosensor in mitochondria (Supplementary Fig. 15a). Exogenous L-2-HG also induced an increase in the emission ratio of mitochondrial LHGFR_ON3C (Supplementary Fig. 15b). The emission ratio of LHGFR_ON3C localized in mitochondria was similar to that of in cytosol (Supplementary Fig. 15c). The uniform distribution of L-2-HG confirmed the presence of a transporter responding for the transport of L-2-HG between cytosol and mitochondria.

HEK293FT cells were then co-transfected with siRNA targeting L2HGDH and LHGFR_ON3C. As shown in Fig. 6e, the transfection of siRNA targeting L2HGDH increased the emission ratio of LHGFR_ON3C, indicating an accumulation of intracellular L-2-HG due to a decrease in L2HGDH levels. Cells with different emission

ratios could be easily distinguished in a mixture of HEK293FT cells with or without L2HGDH knockdown (Supplementary Fig. 16a–b). As expected, overexpression of L2HGDH decreased the emission ratio of LHGFR_ON3C, further supporting the function of L2HGDH in L-2-HG catabolism (Supplementary Fig. 16c).

The response of LHGFR_ON3C to changes in hypoxia-induced production of L-2-HG in human cells was also studied. The emission ratio of LHGFR_ON3C in HEK293FT cells after 24 h exposure to 2% oxygen was higher than the ratio obtained under normoxic conditions and hypoxia induced a 3.5-fold increase in the concentration of L-2-HG (Fig. 6f). In addition, exogenous cell-permeable dimethyl-2-ketoglutarate significantly increased the emission ratio of LHGFR_ON3C under hypoxic conditions, suggesting that hypoxia-induced L-2-HG might originate from 2-KG (Fig. 6f).

LDHA and MDH2 have been reported to participate in hypoxia-induced L-2-HG production due to their "promiscuous" catalytic activities[1–4]. In support of this conclusion, siRNAs targeting LDHA and MDH2 were transfected separately or in combination into LHGFR_ON3C-expressing HEK293FT cells. As shown in Fig. 6g, the decrease of LDHA and MDH2 reduced the emission ratio of LHGFR_ON3C under hypoxic conditions, suggesting that these enzymes indeed contribute to the production of L-2-HG from 2-KG.

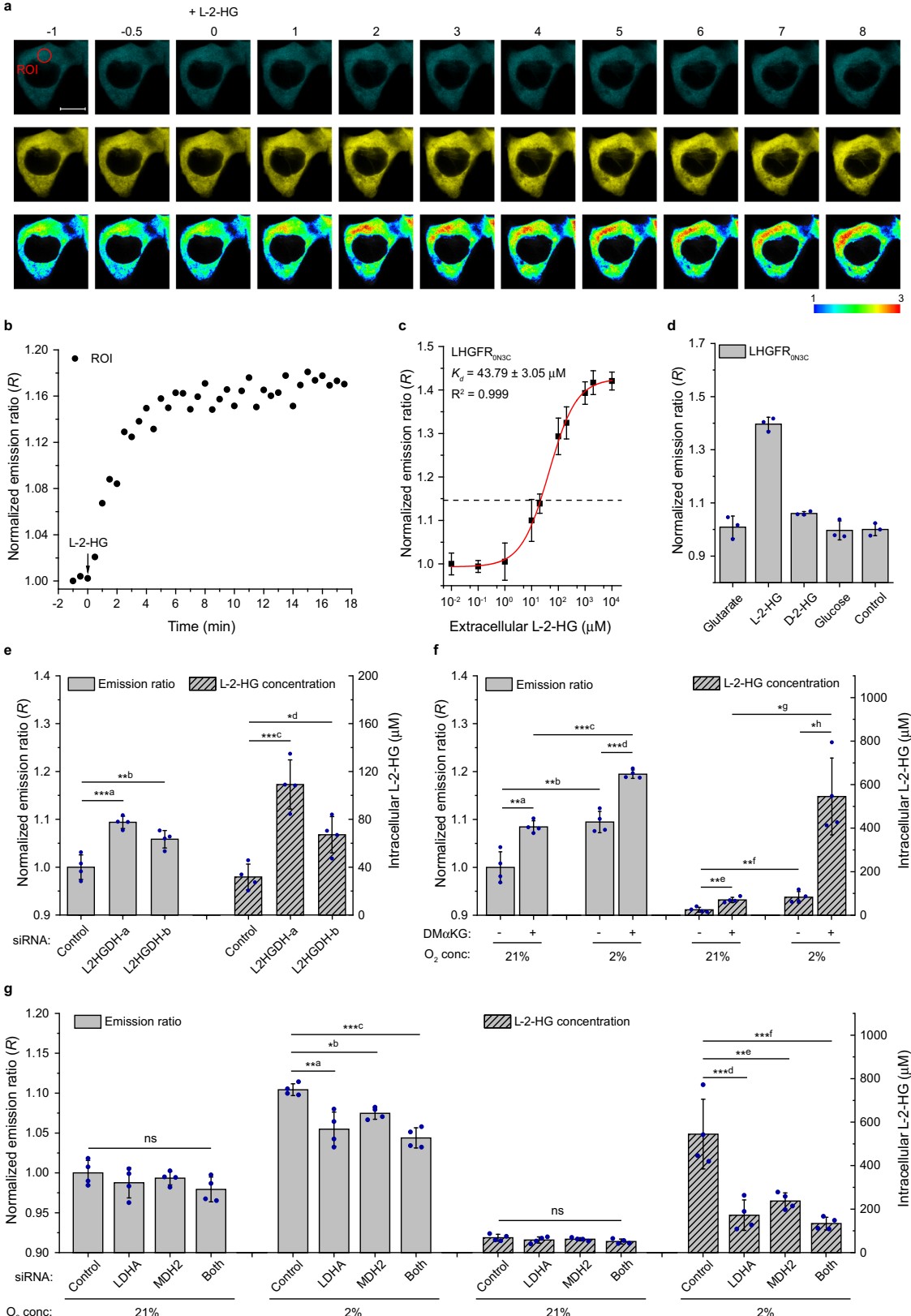

The performance of LHGFR$_{0N7C}$ in monitoring L-2-HG fluctuations in living HEK293FT cells was also studied and similar results were acquired (Supplementary Fig. 17). In addition, the emission ratio of control cells expressing LHGFR$_{3N7C}$ remained unaffected by exogenous L-2-HG addition and hypoxia treatment (Supplementary Fig. 18).

## Discussion

Bacteria have evolved to respond and catabolize a wide range of metabolites. The availability of genomic information from different organisms provided researchers with a new route to predict transcriptional regulators and their physiological functions. In this study, we used a genomic neighborhood analysis approach

**Fig. 6 Monitoring L-2-HG fluctuations in human cells by LHGFR$_{0N3C}$. a** Sequential images of mTFP (top), Venus (middle), and Venus/mTFP emission ratio (bottom, pseudocolored) of single HEK293FT cell expressing LHGFR$_{0N3C}$. 10 mM L-2-HG was added at time point zero (min). Elapsed time (in minutes) after the addition of L-2-HG is shown at the top of the images. Scale bar, 10 μm. **b** Time course of the emission ratio changes inside the region of interest (ROI) depicted from the top-left image of **a**. **c** Normalized dose-response curve of LHGFR$_{0N3C}$ expressed in HEK293FT cells with increasing concentrations (10 nM to 10 mM) of L-2-HG. Cells were permeabilized with 10 μM digitonin. The emission ratio of non-permeabilized HEK293FT cells under physiological conditions is indicated with a black dash line. **d** Responses of LHGFR$_{0N3C}$ expressed in HEK293FT cells to exogenously added 1 mM glutarate, L-2-HG, D-2-HG, and glucose. Cells were permeabilized with 10 μM digitonin. All data were normalized to the control (ratio in the absence of any tested compounds). **e** Identification of the function of L2HGDH in L-2-HG catabolism by LHGFR$_{0N3C}$. The emission ratio was measured after co-transfecting siRNA targeting L2HGDH and LHGFR$_{0N3C}$ for 48 h. ***a: $P = 0.0002$, **b: $P = 0.0058$, ***c: $P = 0.0002$, *d: $P = 0.0303$. **f** Detection of hypoxia-induced L-2-HG accumulation by LHGFR$_{0N3C}$. Emission ratio changes were recorded after LHGFR$_{0N3C}$-expressing HEK293FT cells cultured in normoxia or hypoxia in the absence and presence of 5 mM dimethyl-2-ketoglutarate (DMαKG) for 24 h. The emission ratio was normalized to normoxic conditions without DMαKG. **a: $P = 0.0029$, **b: $P = 0.0030$, ***c: $P < 0.0001$, ***d: $P = 0.0002$, **e: $P = 0.0016$, **f: $P = 0.0068$, *g: $P = 0.0126$, *h: $P = 0.0139$. **g** Identification of the functions of LDHA and MDH2 in L-2-HG anabolism by LHGFR$_{0N3C}$. HEK293FT cells were cultured in the presence of 5 mM DMαKG. The emission ratio was normalized to the normoxic conditions treated with negative siRNA. **a: $P = 0.0011$, *b: $P = 0.0400$, ***c: $P = 0.0002$, ***d: $P = 0.0004$, **e: $P = 0.0021$, ***f: $P = 0.0002$. Inconsistent initial emission ratios were detected in HEK293FT cells under different conditions. The normalized emission ratios were thus used to monitor the changes of L-2-HG in different HEK293FT cells. All data shown are means ± s.d. ($n = 3, 3, 4, 4,$ and 4 independent experiments for **c**, **d**, **e**, **f**, and **g**). *$P < 0.05$; **$P < 0.01$; ***$P < 0.001$; ns, no significant difference ($P ≥ 0.05$); one-way ANOVA test with Tukey's Multiple Comparison Test for **e** and **g**; two-tailed $t$-test for **f**.

combined with genetic and biochemical techniques to discover the transcriptional regulator of L-2-HG catabolism. The identified transcriptional regulator, LhgR, was present directly upstream of LhgO in *P. putida* W619 (Fig. 1a). It specifically binds to the promoter region of *lhgO* and represses the transcription of *lhgO* gene. L-2-HG interferes with the DNA-binding activity of LhgR and induces expression of LhgO. LhgR is an example of the bacterial allosteric transcription factor that specially responds to L-2-HG. This finding showed the application of a collection of sequenced genomes in the identification of transcriptional regulators and the approach can be expanded to target other transcriptional regulators in diverse bacteria.

L-2-HG is a harbinger of altered metabolism and participates in the pathogenesis of L-2-hydroxyglutaric acidurias and cancer[12–15,38,39]. Standard methods to measure L-2-HG are based on MS techniques that are time-consuming and require highly skilled workers. LhgR can bind to L-2-HG and then undergo conformational changes, which in turn affects DNA-binding. Thus, a FRET biosensor, LHGFR$_{0N0C}$, utilizing the allosteric transcription factor LhgR as an L-2-HG biorecognition element was constructed for a convenient assay of L-2-HG concentrations. As a ratiometric sensor, the emission ratio changes of LHGFR are not affected by the amount of sensor in biological samples or in living cells, and thus allows more accurate measurements. The biosensor was optimized by truncating the N-terminal and C-terminal domains of LhgR or by adding an artificial linker to the N-terminal and C-terminal regions of LhgR. The optimized variants, LHGFR$_{0N3C}$ and LHGFR$_{0N7C}$, increased $\Delta R_{max}$ from 11.47 ± 0.38% to 56.13 ± 0.29% and 60.37 ± 1.30%, respectively, with significantly improved sensitivity for L-2-HG detection (Fig. 3b, Supplementary Fig. 6a and Fig. 6e). Besides signal recognition, signal transduction is also an essential aspect of the development of biosensors. Various biosensing systems, including CRISPR-Cas12a- and allosteric transcription factors-mediated small molecule detector (CaT-SMelor)[40], allosteric transcription factors-based nicked DNA-template-assisted signal transduction (aTF-NAST)[41], and quantum-dot-allosteric transcription factors-FRET[42], have been reported. Other biosensors based on L-2-HG responding LhgR and the corresponding transduction mechanisms could also be developed for the detection of L-2-HG.

L-2-HG is a biomarker for L-2-hydroxyglutaric aciduria and a variety of cancers. The rapid, sensitive, and specific measurement of L-2-HG in body fluids is of clinical significance[12–15,38,43]. The LODs of LHGFR$_{0N3C}$ and LHGFR$_{0N7C}$ for L-2-HG in serum and urine were 5.84 μM and 15.74 μM, 1.68 μM and 0.92 μM,

respectively (Supplementary Table 3). The reported concentration of L-2-HG in plasma of patients with L-2-hydroxyglutaric aciduria and L-2-HG-associated brain malignancies is about 7–84 μM[38,43]. Thus, these biosensors especially LHGFR$_{0N7C}$ are suitable for the measurement of the endogenous L-2-HG in body fluids of patients with L-2-HG related diseases. D-2-HG concentration in the urine, plasma, and cerebrospinal fluid of patients with combined D,L-2-hydroxyglutaric aciduria or D2HGDH mutation-associated D-2-hydroxyglutaric aciduria was reported to be less than 100 μM[38]. Serum 2-HG in IDH1-mutated and IDH2-mutated cancers like acute myeloid leukemia is about 300 μM[44]. LHGFR$_{0N3C}$ has high LODs for D-2-HG in serum and urine (781.90 μM and 3876.40 μM), which can prevent the false-positive results in patients with D-2-hydroxyglutaric aciduria and/or IDH-mutant cancers (Supplementary Table 4). Compared with LHGFR$_{0N3C}$, LHGFR$_{0N7C}$ has a higher affinity with D-2-HG. However, this biosensor is a sensitive variant with low LOD for L-2-HG (Supplementary Table 3). Assessment of L-2-HG by LHGFR$_{0N7C}$ can be easily conducted after a simple dilution to prevent the possible interference induced by D-2-HG-related diseases. Besides higher sensitivity and specificity, LHGFR$_{0N3C}$ and LHGFR$_{0N7C}$ also have superior accuracy and precision over previous MS-based methods for L-2-HG detection (Fig. 4 and Supplementary Table 2). Being genetically encoded, LHGFR$_{0N3C}$ and LHGFR$_{0N7C}$ could be produced in great quantities by recombinant bacteria with low cost and could be applied in future rapid and sensitive clinical diagnosis of L-2-HG-related diseases.

L-2-HG plays important roles in diverse physiological processes such as hypoxic adaption, immunity, and tumorigenesis, and the establishment of the intracellular detection method of this metabolite is of great research significance[3,12,14,16]. Intracellular L-2-HG concentrations of activated CD8$^+$ T-lymphocytes, renal cancer cells, and various cells under hypoxic conditions were about 25 μM to several hundred micromoles[3,12,16]. Compared with LHGFR$_{0N7C}$, LHGFR$_{0N3C}$ with a higher $K_d$ for L-2-HG (29.33 ± 1.24 μM) is a more viable alternative for the detection of intracellular L-2-HG (Supplementary Fig. 6a and Fig. 6e, Supplementary Table 1). Based on the in vivo response curves of LHGFR$_{0N3C}$, the basal L-2-HG concentration in HEK293FT cells under physiological conditions was determined to be 22.95 ± 11.22 μM (Fig. 6c), which is similar to results acquired using MS-based approaches after complicated sample handling and data analysis[2,3]. LHGFR$_{0N3C}$ also has a low affinity with D-2-HG. The LOD of LHGFR$_{0N3C}$ for D-2-HG was 872.59 μM and the $K_d$ value could not be determined accurately because of the relatively low

affinity (Supplementary Fig. 9c and Supplementary Table 1). D-2-HG at a concentration of 240 μM, which is much higher than its basal intracellular concentrations, could barely affect the response of LHGFR$_{0N3C}$ to L-2-HG (Supplementary Fig. 9e). IDH1-R132H mutation can lead to an extreme intracellular D-2-HG accumulation at millimolar levels[45]. The expression of IDH1-R132H resulted in a 4.9% increase in the emission ratio of LHGFR$_{0N3C}$ under extreme conditions (Supplementary Fig. 19). Calibrated dose-response curves of LHGFR$_{0N3C}$ for L-2-HG would be required for monitoring the fluctuations of L-2-HG in cells with IDH mutations.

The potential of LHGFR for real-time monitoring of fluctuations in intracellular L-2-HG concentrations was illustrated by using bacterial cells and HEK293FT cells. It was revealed that carbon starvation also induced temporary intracellular accumulation of L-2-HG in E. coli cells. CsiD and LhgO played indispensable roles in endogenous anabolism and catabolism of L-2-HG, respectively (Fig. 5f and Supplementary Fig. 13f). In addition, it was identified that the growth of the strain containing lhgO mutation was inhibited when high levels of L-2-HG were present (Fig. 4j-l). Besides being a pathogenic metabolite inducing various cancers and L-2-hydroxyglutaric aciduria in humans[12–15,38,43], L-2-HG would also be a toxic metabolite to bacterial cells. L-2-HG catabolizing enzymes, including L2HGDH in humans[9], dL2HGDH in Drosophila[17], and LhgO in P. putida[7,8] and E. coli[6,46], might all exist as detoxification proteins of L-2-HG. The functions of L2HGDH in L-2-HG catabolism and LDHA and MDH2-mediated 2-KG reduction in hypoxia-induced L-2-HG production were also confirmed in HEK293FT cells using LHGFR as an indicator of L-2-HG. LHGFR can be added to the emerging list of metabolite sensors that have been established in mammalian cells, such as probes for ATP[47], acetylcholine[48], glycine[24], and NAD$^+$/NADH[49]. Several genetically encoded fluorescent metabolite sensors, like the NAD$^+$/NADH probe SoNar, have been successfully applied in the screening of anti-tumor agents[49]. L-2-HG has been exploited as a potential therapeutic target in renal cancer[14] or a biomarker for cancer diagnosis and prognostic assessment[15,50]. The L-2-HG biosensors might also be utilized in the diagnosis and screening of anti-tumor agents for L-2-HG-related cancer.

In summary, a regulatory protein LhgR, which is involved in L-2-HG catabolism and specifically responds to L-2-HG, was identified in P. putida W619. Two LhgR-based L-2-HG biosensors, LHGFR$_{0N3C}$ and LHGFR$_{0N7C}$, with high sensitivity, specificity, and stability, were then constructed. The methods for quantitative estimation of L-2-HG concentrations in various biological samples and living cells by using L-2-HG biosensors were also established. We expect these LhgR-based L-2-HG biosensors to be of practical interest in future research on the metabolism of L-2-HG and the diagnosis and treatment of L-2-HG-related diseases.

## Methods

**Bacterial strains and culture conditions**. The bacterial strains used in this study are listed in Supplementary Data 1. E. coli and its derivatives were cultured in Luria–Bertani (LB) broth at 37 °C and 180 rpm. P. putida KT2440 and its derivatives were grown in minimal salt mediums (MSMs) containing different carbon sources at 30 °C and 200 rpm. Antibiotics were used at the following concentrations: tetracycline at 30 μg mL$^{-1}$; kanamycin at 50 μg mL$^{-1}$; ampicillin at 100 μg mL$^{-1}$; spectinomycin, at 50 μg mL$^{-1}$; and chloramphenicol at 40 μg mL$^{-1}$.

**Cloning of F2-*lhgR*-F1-*lhgO* and *lhgO***. All the plasmids and primers used in this study are listed in Supplementary Data 1 and Data 2, respectively. The gene segment F2-lhgR-F1-lhgO of P. putida W619 was synthesized by Tongyong Biosystem Co., Ltd (China). The lhgO gene of P. putida W619 was amplified and cloned into pME6032 plasmid using the restriction sites of EcoRI and KpnI to construct pME6032-lhgO, and the P$_{tac}$ promoter of pME6032 was replaced by the gene segment F2-lhgR-F1-lhgO using the restriction sites of SacI and BamHI to

construct pME6032-F2-lhgR-F1-lhgO, then both recombinant plasmids were transferred into different derivatives of P. putida KT2440 by electroporation, respectively.

**Construction of *P. putida* KT2440 and *E. coli* MG1655(DE3) mutants**. Genes of P. putida KT2440 were deleted via allele exchange using the pK18 mobsacB system[51]. Briefly, the homologous arms upstream and downstream of the target gene were PCR amplified and fused together by recombinant PCR. The generated fusion fragment was cloned into the suicide plasmid pK18mobsacB. The resulting plasmid was transferred into P. putida KT2440 by electroporation. The single crossover cells and the second crossover cells were sequentially screened from LB plates containing 50 μg mL$^{-1}$ kanamycin or 10% (wt/vol) sucrose, respectively.

To construct the E. coli MG1655(DE3) (ΔcsiD) mutant strain, the homologous arm upstream of the csiD gene, kanamycin resistance cassette, and the homologous arm downstream of the csiD gene were PCR amplified using the primers csiD-F1/csiD-R1, csiD-F2/csiD-R2, and csiD-F3/csiD-R3, respectively. The PCR products were fused together by recombinant PCR, and the resulting fusion was transferred into E. coli MG1655(DE3) harboring pTKRed plasmid following isopropyl-β-D-1-thiogalactopyranoside (IPTG) induction. The recombinant cells were selected on LB plates containing 50 μg mL$^{-1}$ kanamycin at 37 °C. The pCP20 plasmid was transferred into the selected cells, followed by a second screening on LB plates containing 40 μg mL$^{-1}$ chloramphenicol at 30 °C, then cultured in LB medium at 42 °C to eliminate pCP20 plasmid. The lhgO mutant of E. coli MG1655(DE3) was generated by the same process. All mutants were verified by PCR and sequencing.

**Enzymatic assay of LhgO**. The derivatives of P. putida KT2440 were cultured in 50 mL MSMs with 5 g L$^{-1}$ different compounds as carbon sources at 30 °C and 200 rpm. The cells were harvested at mid-log phase, washed twice and resuspended in phosphate-buffered saline (PBS), then lysed by sonication on ice after the addition of 1 mM phenylmethylsulfonyl fluoride (PMSF). The supernatants obtained were used for further enzyme activity measurements after a centrifugation process (13,000 × g for 10 min at 4 °C). Protein concentrations of the supernatants were determined using the Bradford protein assay kit (Sangon, China).

The activity of LhgO was assayed at 30 °C by monitoring the reduction of dichlorophenol-indophenol (DCPIP) corresponding to the change of absorbance at 600 nm using a UV/visible spectrophotometer (Ultrospec 2100 pro, Amersham Biosciences, USA). The 800 μL reaction solution contained 0.1 mM L-2-HG, 0.05 mM DCPIP, 0.2 mM phenazine methosulfate (PMS) in PBS and 40 μL crude extracts. One unit of LhgO activity was defined as the amount of enzyme that catalyzed the reduction of 1 μmol of DCPIP per minute.

**Expression, purification, and characterization of LhgR**. To express and purify the recombinant LhgR, the lhgR gene was PCR amplified using the primer pair lhgR-F/lhgR-R, which contained BamHI and HindIII restriction sites, respectively, and then cloned into the pETDuet-1 plasmid to construct pETDuet-lhgR. The E. coli BL21(DE3) strains harboring pETDuet-lhgR plasmid were grown to an OD$_{600}$ of 0.6 in LB medium at 37 °C, after which the cells were induced for 12 h with 1 mM IPTG at 16 °C. The cells were harvested, washed twice, and resuspended in buffer A (20 mM sodium phosphate and 500 mM sodium chloride, pH 7.4), then lysed by sonication on ice after the addition of 1 mM PMSF and 10% (vol/vol) glycerol. The cell lysate was centrifuged at 13,000 × g for 40 min at 4 °C, and the resultant supernatant was loaded onto a HisTrap HP column (5 mL) equilibrated with buffer A. The target protein was eluted with buffer B (20 mM sodium phosphate, 500 mM sodium chloride, and 500 mM imidazole, pH 7.4), analyzed by 12.5% sodium dodecyl sulfate-polyacrylamide gel electrophoresis (SDS-PAGE), and quantified by the Bradford protein assay kit (Sangon, China).

To determine the native molecular weight of LhgR, gel-filtration chromatography was performed using a Superdex 200 10/300 GL column (GE Healthcare, USA) and standard proteins including thyroglobulin (669 kDa), ferritin (440 kDa), aldolase (158 kDa), conalbumin (75 kDa), ovalbumin (43 kDa), and ribonuclease A (13.7 kDa). The eluent buffer contained 50 mM sodium phosphate and 150 mM sodium chloride (pH 7.2).

**Electrophoretic mobility shift assays**. Electrophoretic mobility shift assays (EMSAs) were carried out using the DNA fragment (F1 or F2) and purified LhgR. The DNA fragments were first amplified by primer pairs F1-F/F1-R and F2-F/F2-R, respectively. Then, either DNA fragment at a concentration of 10 nM DNA was incubated with LhgR (0–160 nM) in 20 μL EMSA binding buffer (10 mM Tris-HCl [pH 7.4], 50 mM KCl, 0.5 mM EDTA, 10% [vol/vol] glycerol, and 1 mM dithiothreitol [DTT]). The binding reactions were carried out at 30 °C for 30 min. Electrophoresis was performed on 6% native polyacrylamide gels at 4 °C and 170 V (constant voltage) for about 45 min, followed by staining with SYBR green I (TaKaRa, China) and photographing. Analysis of the interaction between lhgO promoter region (F1) and LHGFR was performed using the same procedure.

To characterize the effector of LhgR, purified LhgR was first incubated with L-lysine, 5-aminovalerate, glutarate, L-2-HG, D-2-HG, 2-KG, or succinate at 30 °C for 15 min, followed by incubation with the added DNA fragments at 30 °C for 30 min. The mixtures were subsequently subjected to electrophoresis.

**DNase I footprinting**. DNase I footprinting assays were performed using the 6-carboxyfluorescein (FAM) labeled probe and purified LhgR. The DNA fragment F1 was PCR amplified using the primer pair F1-F/F1-R. The PCR products were cloned into the pEASY-Blunt plasmid using pEASY-Blunt Cloning Kit (TransGen, China). The FAM-labeled probes were PCR amplified using the resulting plasmid and the primer pair M13F-FAM/M13R. Then, 350 ng probes were incubated with 2 μg purified LhgR in a total volume of 40 μL for 30 min at 30 °C. The DNase I digestion reaction was carried out by adding a total volume of 10 μL solution containing approximately 0.015 units of DNase I (Promega, USA) and 100 nmol CaCl2 and further incubating for 1 min at 37 °C, then stopped by adding a total volume of 140 μL stop solution containing 0.15% (wt/vol) SDS, 200 mM unbuffered sodium acetate, and 30 mM EDTA. The digested DNA fragments were first extracted with phenol-chloroform, then precipitated with ethanol and resuspended in 30 μL MiliQ water. The binding region of LhgR to DNA fragment F2 was analyzed using the same procedure.

**Construction and purification of LHGFR**. The genes encoding mTFP and Venus were synthesized by Tongyong Biosystem Co., Ltd (China). The mTFP gene and Venus gene were amplified and cloned into pETDuet-1 plasmid using the BamHI and SacI restriction sites, and SalI and NotI restriction sites, respectively. Then either the full-length *lhgR* gene, its truncated variants, or variants with artificial linkers were inserted between mTFP and Venus by the T5 exonuclease DNA assembly (TEDA) method[52], respectively. The L-2-HG biosensor LHGFR and its derivatives were expressed and purified using the same procedure. For expression in HEK293FT cells, the codon-optimized LHGFR$_{0N3C}$, LHGFR$_{0N7C}$, or LHGFR$_{3N7C}$ sequence was synthesized and cloned into pcDNA3.1$^{(+)}$ plasmid behind a Kozak sequence, 5′-GCCACC-3′. To construct the plasmid for mitochondrial expression of LHGFR, the gene of mitochondrial targeting sequence (MLSLRQSIRFFKPATRTLCSSRYLL) was PCR amplified using primer pair Mito-F/Mito-R, and primer pair Mito-LHGFR-F/Mito-LHGFR-R was used to amplify the LHGFR fragment. The products were assembled using overlap PCR with Mito-F/Mito-LHGFR-R, and cloned into pcDNA3.1$^{(+)}$ plasmid behind a Kozak sequence using the NheI and NotI restriction sites.

**Characterization of LHGFR in vitro**. Purified L-2-HG biosensors and different compounds were diluted by 50 mM Tris-HCl buffer (pH 7.4), mixed together in a black 96-well plate at a volume ratio of 3:1, and the fluorescence intensities were measured using an EnSight microplate reader (PerkinElmer, USA) with excitation at 430 nm, emission at 485 nm (mTFP) and 528 nm (Venus). The dose-response curves were fitted by OriginPro 2016 software (OriginLab) according to the following formula:

$$R = R_{\min} + \frac{R_{\min} - R_{\max}}{1 + ([\text{L-2-HG}]/K_d)^p} \quad (1)$$

where $R$, $R_{\min}$, and $R_{\max}$ refer to the emission ratio of Venus to mTFP, ratio in the absence of L-2-HG, and ratio at saturation with L-2-HG, respectively. The [L-2-HG], $K_d$, and $p$ refer to the L-2-HG concentration, apparent dissociation constant, and Hill slope, respectively. Emission spectra were recorded at 430 nm excitation, in steps of 2 nm. Excitation spectra were recorded using emission at 550 nm, excitation from 380 to 535 nm in steps of 2 nm.

Kinetics of L-2-HG binding by LHGFR were assessed by using SX-20 stopped-flow fluorimeter (Applied Photophysics, UK). Equal volumes of 1 μM purified LHGFR in 50 mM Tris-HCl buffer (pH 7.4) and L-2-HG-containing buffer (varying concentrations) were mixed, with data detected every four milliseconds at 430 nm excitation. Emission was detected by using a photomultiplier and a 515 nm long-pass filter, and the detector voltage was set to 400 V. Apparent rate constants ($k^{app}$ = $k_{on}$[L-2-HG] + $k_{off}$) determined by fitting the Venus fluorescence increase after L-2-HG addition with a single exponential equation were plotted against L-2-HG concentrations ([L-2-HG]). Effects of temperature on LHGFR were detected by analyzing the dose-response curves for L-2-HG at 25, 28, 31, 34, 37, 40, and 45 °C, respectively. The reversibility of LHGFR was determined by recording the emission ratios every minute after the addition of 5 μM purified LhgO, the control test without the addition of L-2-HG or purified LhgO was run in parallel. The pH stability of LHGFR was determined using 50 mM Tris-HCl buffer with pH adjusted from 4.0 to 9.0. The background fluorescence without the addition of LHGFR was subtracted.

In order to test the ability of LHGFR for quantitative analysis L-2-HG in different biological samples, purified LHGFR was diluted by 50 mM Tris-HCl (pH 7.4), while varying concentrations of L-2-HG were added into the serum and urine of a healthy adult and bacteria culture medium and filtered through a 0.22 μm filter, respectively. The serum and urine were collected from a healthy adult (the first author of this article). The blood was collected by using promoting coagulating tubes and venous blood collection method, placed for 2 h at room temperature, and serum was prepared by centrifugation for 10 min at 2000 × *g* and 4 °C. The urine collected from the experiment operator was diluted with an equal volume of 50 mM Tris-HCl buffer (pH 7.4). The processed serum and urine were filtered through a 0.22 μm filter and stored at −20 °C until L-2-HG addition. The mixtures of purified LHGFR and L-2-HG in different biological samples were then incubated in a black 96-well plate at a volume ratio of 3:1, and the emission ratios were determined using an EnSight microplate reader. The background fluorescence

without the addition of LHGFR was subtracted. The formulas for the quantitative analysis of L-2-HG in different biological samples by LHGFR$_{0N3C}$ and LHGFR$_{0N7C}$ were as follows:

$$[\text{L-2-HG}](\mu M) = 26.20974 \times \left( \frac{0.79781}{2.17827 - R} - 1 \right)^{1.1705} \quad (2)$$

where $R$ refers to the emission ratio of Venus to mTFP determined by LHGFR$_{0N3C}$, and

$$[\text{L-2-HG}](\mu M) = 7.71913 \times \left( \frac{0.99281}{2.67083 - R} - 1 \right)^{1.0787} \quad (3)$$

where $R$ refers to the emission ratio of Venus to mTFP determined by LHGFR$_{0N7C}$.

**Quantification of L-2-HG by HPLC and LC-MS/MS**. When L-2-HG was used as a carbon source to cultivate *P. putida* KT2440, its consumption was analyzed by using high-performance liquid chromatography (HPLC) system (Agilent 1100 series, Agilent Technologies, USA) equipped with an Aminex HPX-87H column (300 × 7.8 mm, Bio-Rad, USA) and a RID detector at 55 °C. The mobile phase was 0.1% formic acid at a flow rate of 0.4 mL min$^{-1}$.

To detect L-2-HG concentrations in various biological samples by liquid chromatography-tandem mass spectrometry (LC-MS/MS) system, the samples containing D,L-2-hydroxyglutarate disodium salt (2,3,3-D3) as internal standard (ITSD) were centrifuged at 13,000 × *g* for 15 min, then filtered through a 0.22 μm filter. The serum samples were mixed with methanol at a volume ratio of 1:3 and vortexed for 2 min to remove protein before centrifugation. Samples were analyzed by using a Thermo ultimate 3000 rapid separation liquid chromatography system (ThermoFisher, USA) coupled with a Bruker impact HD ESI-Q-TOF mass spectrometer (Bruker Daltonics, Germany) in negative ion mode and equipped with a Chirobiotic R column (250 × 4.6 mm, Supelco Analytical, USA). The mobile phase was prepared from (A) 0.1% triethylamine adjusted to pH 4.5 with acetic acid or (B) methanol. The quantification was conducted with an injection volume of 20 μL, a constant 5% gradient of (B) at a flow rate of 0.5 mL min$^{-1}$, and a total analysis time of 15 min.

**Characterization of LHGFR in living bacteria**. *E. coli* BL21(DE3) strains harboring either pETDuet-LHGFR$_{0N3C}$, pETDuet-LHGFR$_{0N7C}$, or pETDuet-LHGFR$_{3N7C}$ were grown to an OD$_{600}$ of 0.6 in LB medium at 37 °C, after which the cells were induced overnight in the presence of 1 mM IPTG at 16 °C. The cultures were collected by centrifugation at 6000 × *g* for 5 min, washed three times, and resuspended to an OD$_{600}$ of 2.5 by carbon starvation medium (MSM containing no carbon source) or glucose medium (MSM containing 20 mM glucose).

To characterize the sensitivity and specificity of LHGFR expressed in *E. coli* BL21(DE3), 90 μL cell suspensions following 8 h carbon starvation were mixed with 10 μL increasing concentrations of L-2-HG or other compounds, and then added into a black 96-well plate (total 100 μL/per well), the fluorescence intensities were determined using an EnSight microplate reader (PerkinElmer, USA) and the following instrument settings: excitation at 430 nm, emission at 485 nm (mTFP) and 528 nm (Venus), time intervals of 5 min, the temperature at 37 °C, and shake at 180 rpm. For carbon starvation experiments, cell suspensions in carbon starvation medium or glucose medium were added into a black 96-well plate (100 μL/per well), then the fluorescence intensities were monitored every ten minutes. In order to analyze functions of CsiD and LhgO in endogenous L-2-HG anabolism and catabolism during carbon starvation, pETDuet-LHGFR$_{0N3C}$, pETDuet-LHGFR$_{0N7C}$, or pETDuet-LHGFR$_{3N7C}$ was transferred into *E. coli* MG1655(DE3) and its variants, and the assays were performed using the same procedure.

**Cell culture and live-cell imaging**. HEK293FT cells were cultured in high-glucose Dulbecco's modified eagle medium (DMEM) supplemented with 10% (vol/vol) fetal bovine serum (FBS), 100 units mL$^{-1}$ penicillin, and 100 μg mL$^{-1}$ streptomycin (all purchased from ThermoFisher, USA), and kept at 37 °C in humidified air containing 5% CO2. For hypoxia experiments, cells were kept in a compact O2 and CO2 subchamber controller (ProOx C21, BioSpherix, USA) at 2% O2, 5% CO2, and balanced with N2 for 24 h. For the construction of LHGFR expressing cell, HEK293FT cells were plated in 24-well plates so that they reached 70-90% confluency 24 h after plating, the medium was refreshed 2 h before transfection. Lipoplexes were first prepared in 50 μL Opti-MEM Reduced Serum Medium (ThermoFisher, USA) per well containing 1.5 μL Lipofectamine 3000 (ThermoFisher, USA) and 1 μg pcDNA3.1$^{(+)}$ plasmid encoding either LHGFR$_{0N3C}$, LHGFR$_{0N7C}$, or LHGFR$_{3N7C}$, and incubated for 15 min at room temperature, then added into the cell cultures.

For live-cell imaging, HEK293FT cells were plated on a poly-L-lysine pre-coated 35 mm glass-bottom dish and transfected with LHGFR after 24 h. Live-cell imaging was carried out 48 h following transfection by using a Zeiss 880 confocal microscope equipped with an Objective C-Apochromat ×40/1.2 W autocorr M27 lens, a 458 nm argon laser, and a full-spectrum fluorescence detector. The emission of LHGFR expressed in HEK293FT cells was divided into a 463–495 nm channel (mTFP) and a 505–700 channel (Venus). Images were captured using 800 gain, 1024 × 1024 frame size, and 8 bit depth. The fluorescence intensities of each channel were analyzed in ZEN 3.1 software, and raw data were exported to Image-

Pro Plus software for ratio image analysis. The Venus/mTFP emission ratio was calculated by dividing pixel-by-pixel a Venus image with a mTFP image. To real-time monitor the emission ratio of LHGFR in single living cells, single-cell with moderately fluorescent was imaged every 30 seconds, and 10 mM L-2-HG was added into the cultures when the ratio of two-channel fluorescence intensities was stable.

**Characterization of LHGFR in HEK293FT cells**. To characterize the sensitivity and specificity of LHGFR expressed in HEK293FT cells, cells were trypsinized 48 h following transfection and suspended in 1× Hank's balanced salt solution supplemented with 20 mM HEPES. Increasing concentrations of L-2-HG or other compounds including glutarate, D-2-HG, and glucose was mixed with the cell suspensions in a 96-well plate, respectively. Digitonin at a concentration of 10 μM was used to induce cell permeabilization and deplete intracellular L-2-HG for in vivo response curves construction. Then, the fluorescence intensities were determined by a SpectraMax i3 fluorescence plate reader (Molecular Devices, USA) with excitation at 430 nm and emission at 485 nm (mTFP) and 528 nm (Venus). Basal L-2-HG concentration in HEK293FT cells under physiological conditions was determined by substituting the emission ratios of non-permeabilized HEK293FT cells into the calibrated in vivo response curves.

For the detection of hypoxia-induced production of L-2-HG, LHGFR$_{0N3C}$ or LHGFR$_{3N7C}$ was expressed in HEK293FT cells and cultured sequentially under normoxic conditions for 24 h and hypoxic conditions for 24 h in the absence or presence of 5 mM dimethyl-2-ketoglutarate (DMαKG). The preparation of cell suspensions and the measurement of emission ratios were performed using the same procedure. The background fluorescence was subtracted at each emission wavelength.

**siRNA experiments**. The following Silencer Select siRNAs used in this study were purchased from ThermoFisher Scientific (USA): negative control (4390846), L2HGDH-a (s36692), L2HGDH-b (s36693), LDHA (s351), and MDH2 (s8622). To analyze L2HGDH functions in L-2-HG catabolism, siRNA targeting L2HGDH and pcDNA3.1$^{(+)}$ plasmid encoding either LHGFR$_{0N3C}$, LHGFR$_{0N7C}$, or LHGFR$_{3N7C}$ were mixed with Lipofectamine 3000 Transfection Reagent (ThermoFisher, USA) in Opti-MEM Reduced Serum Medium, and the lipoplexes prepared were transfected into HEK293FT cells according to the manufacturer's protocol. The fluorescence intensities were measured by a SpectraMax i3 fluorescence plate reader 48 h following transfection. Similarly, HEK293FT cells were transfected by LHGFR$_{0N3C}$ or LHGFR$_{3N7C}$ and siRNAs targeting LDHA and MDH2 separately or in combination. After transfection, cells were cultured sequentially under normoxic conditions for 24 h and hypoxic conditions for 24 h in the presence of 5 mM DMαKG, then the fluorescence intensities were measured. The cells cultured under the normoxic conditions in the presence of 5 mM DMαKG for 48 h were set as control. The background fluorescence was subtracted at each emission wavelength.

**Statistics and reproducibility**. Software for initial data processing was Microsoft Excel 2016, and subsequent analyses were carried out using OriginPro 2016 (OriginLab), OriginPro 2019 (OriginLab), Graphpad Prism 5 (Graphpad), and Graphpad Prism 7 (Graphpad). The fluorescence intensities were determined by using Kaleido 3.0 (PerkinElmer) and SoftMax Pro software 7.0.2 (Molecular Devices). The imaging data were obtained and processed by Zen 3.1 (Zeiss) and Image-Pro Plus 6.0. All data shown are means ± s.d. and were analyzed using one-way ANOVA test with Tukey's Multiple Comparison Test or two-tailed $t$-test where appropriate; *$P < 0.05$; **$P < 0.01$; ***$P < 0.001$; ns, no significant difference ($P \geq 0.05$). For SDS-PAGE analyses, EMSAs, and fluorescence imaging experiments, similar results were obtained from three independent experiments. Detailed data analyses are described in the text.

**Reporting summary**. Further information on research design is available in the Nature Research Reporting Summary linked to this article.

## Data availability
The data supporting the findings of this study are available within the article, its Supplementary Information files, and the Source Data file provided with this paper. A reporting summary for this article is available as a Supplementary Information file. Source data are provided with this paper.

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

## Acknowledgements

This work was supported by the grants of the National Key R&D Program of China (2018YFA0901200), National Natural Science Foundation of China (31970055), Shandong Provincial Funds for Distinguished Young Scientists (JQ 201806), and Qilu Young Scholar of Shandong University. The funders had no role in study design, data collection, and interpretation, or the decision to submit the work for publication. We also thank Dr. Zhifeng Li, Dr. Jingyao Qu, and Dr. Jing Zhu from SKLMT (State Key Laboratory of Microbial Technology, Shandong University) for assistance in the mass spectrographic analysis, Dr. Haiyan Yu, Dr. Yuyu Guo, and Dr. Xiaomin Zhao from SKLMT for assistance in microimaging of laser scanning confocal microscope analysis.

## Author contributions

C.G., C.M., and P.X. designed the research. Z.K., M.Z., K.G., W.M., Y.L., D.X., W.Z., and S.G. performed the research. Z.K. and C.G. analyzed the data. Z.K., C.G., C.M., and P.X. wrote the paper.

## Competing interests
The authors declare no competing interests.
