## [Peer Review File · Nature Communications]

Reviewers' Comments:

Reviewer #1:

Remarks to the Author:

In this manuscript by Kang et al, the authors identify an L-2-hydroxyglutarate (L-2HG) responsive transcriptional repressor dubbed LhgR in a strain of pseudomonas bacteria and engineer/optimize LhgR to produce a FRET-based fluorescent biosensor for L-2HG. The prior manuscript was reviewed as a submission to Nature Chemical Biology. The authors have performed an enormous amount of work to address the comments and suggestions from the previous reviewers. The revised manuscript is much improved and nearly ready for publication. The findings should be of interest to the diverse readership of Nature Communications.

My prior suggestions/concerns have been addressed with substantial new data:

1. Demonstrating lack of response to L-lactate and a wide range of other structurally related metabolites. This helps to bolster confidence in the specificity of the reporter.
2. Behavior of the sensor in cells harboring a D-2HG producing IDH mutation.
3. Response curves to biofluids containing D-2HG.
4. Ability of the sensor to discriminate cells with or without elevated L-2HG from a mixture.
- 5/6. Standard curves used to calculate concentrations of L-2HG are now clear.
7. Explanation of normalized emission ratios is now provided.
8. Emission spectra for the LHGFR0N3C and LHGFR0N7C reporters are now provided.
9. Various problematic wording is now corrected.

There are a few minor suggestions and/or points of clarification which can be addressed with text.

1. How many bacterial strains were found with LhgR? Do the authors know what is source of L-2HG in bacteria like *P. putida* W619 with no *csiD* gene? Can the authors speculate as to why the only strain(s) with LhgR, also lack *csiD*? Is there a presumed biologic reason for this?
2. Are the concentrations in figure Fig 2e correct? 50 mM of L-2HG and other metabolites were used here?
3. Please provide a little more explanation on how the conformational change in LhgR was predicted.
4. For Figure 6, it is not clear which panels showed permeabilized cells, please be sure this is indicated.

Reviewer #2:

Remarks to the Author:

The authors did a great job to revise the manuscript, by adding more details to the methods, more experimental controls data for clarity and more biological significant data of live cells, which solved most of my previous concerns. I therefore recommend the publication of the manuscript.

I have one last suggestion:

Supplementary Figure 10. In addition to plot the Apparent rate constants of the sensor's response, the authors should also provide original time course data of stop flow kinetic measurement for better understanding of the sensors' respond speed.

Reviewer #3:

Remarks to the Author:

The authors discovered LhgR, a transcription factor that binds L-2-hydroxyglutarate (L-2-HG), from the eubacterium *Pseudomonas putida* W619, and this discovery is highly novel because no gene expression regulation system by L-2-HG has been found so far. Furthermore, the authors developed a FRET-type fluorescent biosensor LHGFR that is based on LhgR and specifically reports the concentration of L-2-HG. This is the first fluorescent biosensor for L-2-HG. By comparison with LC/MS data, LHGFR can measure the concentration of L-2-HG in living cells with high quantitative accuracy. In addition, quantitative measurement in biological samples such as urine and blood was also confirmed, and future application to clinical testing is expected. Imaging of L-2-HG in living mammalian cells strongly suggests that the conversion of alpha-KG to L-2-HG is enhanced under hypoxia, indicating that this biosensor is extremely useful for the study of L-2-HG metabolism. Since the physiological functions of L-2-HG have attracted a great deal of attention in recent years, I think that this biosensor will be a very important tool for promoting L-2-HG research. Overall, the manuscript is well written, the experiments appear to be well conducted, and the conclusions seem to be valid.

Minor points

1. Is LHGFR a dimer or a monomer? This is important information when users want to fuse the biosensor to another protein for targeting the biosensor to the specific location of the cell.
2. According to Fig6a, L-2-HG concentration seems to be non-uniform within a HEK293 cell. Is a region with high L-2-HG close to specific organelles, such as mitochondria? Please discuss the non-uniform distribution of L-2-HG.

Responses to the reviewers' comments point-by-point:

Thanks a lot for the reviewers' good comments and thoughtful suggestions, which are very helpful to improve our manuscript. We have carefully evaluated the reviewers' comments and revised our manuscript. With regard to reviewers' comments and suggestions, we reply as follows:

To Reviewer #1 (Remarks to the Author):

In this manuscript by Kang et al, the authors identify an L-2-hydroxyglutarate (L-2HG) responsive transcriptional repressor dubbed LhgR in a strain of pseudomonas bacteria and engineer/optimize LhgR to produce a FRET-based fluorescent biosensor for L-2HG. The prior manuscript was reviewed as a submission to Nature Chemical Biology. The authors have performed an enormous amount of work to address the comments and suggestions from the previous reviewers. The revised manuscript is much improved and nearly ready for publication. The findings should be of interest to the diverse readership of Nature Communications.

My prior suggestions/concerns have been addressed with substantial new data:

1. Demonstrating lack of response to L-lactate and a wide range of other structurally related metabolites. This helps to bolster confidence in the specificity of the reporter.

2. Behavior of the sensor in cells harboring a D-2HG producing IDH mutation.

3. Response curves to biofluids containing D-2HG.

4. Ability of the sensor to discriminate cells with or without elevated L-2HG from a mixture.

5/6. Standard curves used to calculate concentrations of L-2HG are now clear.

7. Explanation of normalized emission ratios is now provided.

8. Emission spectra for the LHGFR0N3C and LHGFR0N7C reporters are now provided.

9. Various problematic wording is now corrected.

There are a few minor suggestions and/or points of clarification which can be addressed with text.

Response: Thanks for your positive comments. We have made the corresponding

revisions according to your suggestions.

1. How many bacterial strains were found with *lhgR*? Do the authors know what is source of L-2-HG in bacteria like *P. putida* W619 with no *csiD* gene? Can the authors speculate as to why the only strain(s) with *lhgR*, also lack *csiD*? Is there a presumed biologic reason for this?

Response: Thanks for your good comments. In this study, bacteria containing LhgO encoding gene *lhgO* were selected to study the regulation of L-2-HG metabolism and to search an L-2-HG specific regulator for biosensor construction. Homologs of LhgO could be found in 612 different bacterial strains, while homologs of CsiD could be found in 454 different bacterial strains (DOI: 10.1038/s41467-018-04513-0). In the bacterial strains with both CsiD and LhgO, such as *Pseudomonas putida* KT2440, LhgO is involved in metabolism of both L-2-HG and gluturate. The *csiD-lhgO* gene cluster is regulated by the allosteric transcription factor CsiR, which uses both L-2-HG and gluturate as its effectors (DOI: 10.1128/mBio.01570-19). We speculated that LhgO might be solely involved in L-2-HG metabolism in the bacterial strains with only LhgO and regulated by an L-2-HG specific allosteric transcription factor. Therefore, we explored the existence of L-2-HG-specific transcription factor in bacteria without CsiD in this work and fortunately found the L-2-HG specific regulator LhgR in *P. putida* W619. Using a BLASTP program in the sequenced bacterial genomes from GenBank, homologs of LhgR can be found in many bacterial strains. However, whether these regulators also specifically response to L-2-HG and regulate LhgO expression needs a case-to-case study. Thus, the number of bacterial strains with LhgR is rather difficult to predict and we cannot directly conclude that all of the strains with LhgR also lack CsiD.

In mammals and plants, L-2-HG is produced by lactate dehydrogenase (LDH) and malate dehydrogenase (MDH)-mediated 2-ketoglutarate (2-KG) reduction. There are MDH encoding genes (PputW619_3915 and PputW619_4438) in genome of *P. putida* W619. We speculated that the intracellular L-2-HG may also be derived from the “promiscuous” catalytic activities of MDH toward 2-KG in *P. putida* W619. In addition, *P. putida* W619, like *P. putida* KT2440, might also use the extracellular L-2-HG as the sole carbon source for

growth. The possible source of L-2-HG in *P. putida* W619 was briefly discussed in the revised manuscript as follows:

“These results suggested that L-2-HG can specifically prevent the binding of LhgR to the promoter of *lhgO* and induce its expression. LhgR may help *P. putida* W619 to specifically sense L-2-HG generated by intracellular metabolism or present in habitats and stimulate the catabolism of L-2-HG.”

2. Are the concentrations in figure Fig 2e correct? 50 mM of L-2HG and other metabolites were used here?

Response: Thanks for your good questions. Yes, in the EMSA experiment in **Fig. 2e**, 50 mM L-2-HG and other metabolites were indeed used. The higher concentration was used to ensure that LhgR could dissociate from the target DNA.

Figure 2 Purification and characterization of LhgR. **(e)** L-2-HG prevent LhgR binding to the *lhgO* promoter region. EMSAs were carried out with F1 fragment (10 nM) and purified LhgR (60 nM) in the absence of any other tested compounds (No ligand) and in the presence of 50 mM different compounds. Lane M was the molecular weight markers; lane 0 without LhgR was used as the control.

3. Please provide a little more explanation on how the conformational change in lhgR was predicted.

Response: Thanks for your good advice. Spectra properties of LHGFR_{0N0C},

LHGFR_{0N3C}, and LHGFR_{0N7C} revealed the addition of L-2-HG could reduce the emission peak at 492 nm of mTFP and increase the emission peak at 526 nm of Venus (Supplementary Fig. 4 and Supplementary Fig. 8), indicating that the L-2-HG binding to LhgR increased FRET between the fluorophores. This result suggests that the relative distance between the fluorescent pair may be shortened, which is due to the conformational change of LhgR in response to L-2-HG binding. On the other hand, the orientation change between the fluorescence pair may also increase the FRET efficiency.

Related discussion was described in the “Results” section as follows:

“Spectra properties of LHGFR_{0N0C} reveal the addition of L-2-HG could reduce the emission peak at 492 nm of mTFP and increase the emission peak at 526 nm of Venus (Supplementary Fig. 4). Thus, the conformational change of LhgR after the L-2-HG binding may lead to a shortened relative distance and/or favorable orientation of mTFP and Venus, resulting in an increase in FRET (Fig. 3a).”

Supplementary Figure 4 Spectra properties of LHGFR_{0N0C}. (a) Fluorescence emission spectrum changes of 1 μM LHGFR_{0N0C} at 430 nm excitation with (red) or without (black) the addition of 100 μM L-2-HG were indicated.

Supplementary Figure 8 Spectra properties of LHGFR_{0N3C} and LHGFR_{0N7C}. **(a, b)** Fluorescence emission spectra changes of 1 μM LHGFR_{0N3C} **(a)** and LHGFR_{0N7C} **(b)** at 430 nm excitation with (red) or without (black) the addition of 100 μM L-2-HG were indicated.

4. For Figure 6, it is not clear which panels showed permeabilized cells, please be sure this is indicated.

Response: Thanks for your good advice. Permeabilized cells were used in **Fig. 6c** and **Fig. 6d**, and the relevant description “Cells were permeabilized with 10 μM digitonin.” was added to the legends of **Fig. 6c** and **Fig. 6d**, respectively.

Figure 6 Monitoring L-2-HG fluctuations in human cells by LHGFR_{0N3C}. **(a)** Sequential images of mTFP (top), Venus (middle), and Venus/mTFP emission ratio (bottom, pseudocolored) of single HEK293FT cell expressing LHGFR_{0N3C}. 10 mM L-2-HG was added at time point zero (min). Elapsed time (in minutes) after addition of L-2-HG is shown at the top of the images. Scale bar, 10 μm . **(b)** Time course of the emission ratio changes inside region of interest (ROI) depicted from the top-left image of **(a)**. **(c)** Normalized dose-response curve of LHGFR_{0N3C} expressed in HEK293FT cells with increasing concentrations (10 nM to 10 mM) of L-2-HG. Cells were permeabilized with 10 μM digitonin. The emission ratio of non-permeabilized HEK293FT cells under physiological conditions is indicated with black dash line. **(d)** Responses of LHGFR_{0N3C} expressed in HEK293FT cells to exogenously added 1 mM glutarate, L-2-HG, D-2-HG, and glucose. Cells were permeabilized with 10 μM digitonin. All data were normalized to the control (ratio in the absence of any tested compounds).

To Reviewer #2 (Remarks to the Author):

The authors did a great job to revise the manuscript, by adding more details to the methods, more experimental controls data for clarity and more biological significant data of live cells, which solved most of my previous concerns. I therefore recommend the publication of the manuscript.

Response: Thank you very much for your time on our paper. We have made the corresponding revisions according to your suggestions.

I have one last suggestion:

Supplementary Figure 10. In addition to plot the Apparent rate constants of the sensor's response, the authors should also provide original time course data of stop flow kinetic measurement for better understanding of the sensors' respond speed.

Response: Thanks for your good suggestions. The time course data of Venus fluorescence intensity changes in response to L-2-HG addition have been added to the revised Supplementary Information as shown below:

Supplementary Figure 10 *In vitro* characterization of purified LHGFR_{0N3C} and LHGFR_{0N7C}. **(a, b)**

Kinetics of L-2-HG binding to LHGFR_{0N3C} **(a)** and LHGFR_{0N7C} **(b)** were measured by using the stopped-flow technique. Different L-2-HG concentrations were mixed with the purified LHGFR_{0N3C} **(a)** and LHGFR_{0N7C} **(b)** protein and the change in Venus fluorescence intensity was measured over time. The Venus fluorescence intensity change fitted by a single exponential equation corresponding to each L-2-HG concentration was shown.

To Reviewer #3 (Remarks to the Author):

The authors discovered LhgR, a transcription factor that binds L-2-hydroxyglutarate (L-2-HG), from the eubacterium Pseudomonas putida W619, and this discovery is highly novel because no gene expression regulation system by L-2-HG has been found so far. Furthermore, the authors developed a FRET-type fluorescent biosensor LHGFR that is based on LhgR and specifically reports the concentration of L-2-HG. This is the first fluorescent biosensor for L-2-HG. By comparison with LC/MS data, LHGFR can measure the concentration of L-2-HG in living cells with high quantitative accuracy. In addition, quantitative measurement in biological samples such as urine and blood was also confirmed, and future application to clinical testing is expected. Imaging of L-2-HG in living mammalian cells strongly suggests that the conversion of alpha-KG to L-2-HG is enhanced under hypoxia, indicating that this biosensor is extremely useful for the study of L-2-HG metabolism. Since the physiological functions of L-2-HG have attracted a great deal of attention in recent years, I think that this biosensor will be a very important tool for promoting L-2-HG research. Overall, the manuscript is well written, the experiments appear to be well conducted, and the conclusions seem to be valid.

Response: Thanks for your positive evaluation of our manuscript and important comments that help us further improve the quality of our work. We have made the corresponding revisions according to your suggestions.

Minor points

1. Is LHGFR a dimer or a monomer? This is important information when users want to fuse the biosensor to another protein for targeting the biosensor to the specific location of the cell.

Response: Thanks for your good advice. The analyses of the native molecular weight of LHGFR_{0N3C} and LHGFR_{0N7C} were added in **Supplementary Fig. 7a-b** (new data). Both LHGFR_{0N3C} and LHGFR_{0N7C} behave as tetramers, and respective results were described in the “Results” section as follows:

“Both LHGFR_{0N3C} and LHGFR_{0N7C} behave as tetramers and have lost the ability to bind DNA (**Supplementary Fig. 7**).”

Supplementary Figure 7 Analysis of the native molecular weight and DNA binding ability of LHGFR_{0N3C} and LHGFR_{0N7C}. **(a, b)** Gel-filtration chromatography of the purified LHGFR_{0N3C} **(a)** and LHGFR_{0N7C} **(b)** with the Superdex 200 10/300 GL column. Red curve, chromatogram of purified LHGFR; Black line, standard curve for protein molecular mass standards.

2. According to Fig6a, L-2-HG concentration seems to be non-uniform within a HEK293 cell. Is a region with high L-2-HG close to specific organelles, such as mitochondria? Please discuss the non-uniform distribution of L-2-HG.

Response: Thanks for your good suggestion. LHGFR_{0N3C} was firstly expressed in the cytosol of HEK293FT cells. As shown in **Fig. 6a**, addition of 10 mM L-2-HG affected mTFP and Venus fluorescence intensities, which caused an immediate and dramatic increase in the emission ratio of LHGFR_{0N3C}-expressing single cell. The non-uniform increase in the emission ratio of LHGFR_{0N3C} might due to the heterogeneous diffusion of extracellular L-2-HG in cytosol.

As for the distribution of L-2-HG in different organelle, L2HGDH, the only reported enzyme that can catabolize L-2-HG in human cells, is localized in mitochondria (DOI: 10.1016/j.trecan.2017.12.005). The mitochondrial targeting sequence was appended to LHGFR_{0N3C} to localize the biosensor in mitochondria (**Supplementary Fig. 15a**). Exogenous L-2-HG also induced increase of emission ratio of mitochondrial LHGFR_{0N3C} (**Supplementary Fig. 15b**). Emission ratio of LHGFR_{0N3C} localized in mitochondria was similar to that of in cytosol (**Supplementary Fig. 15c**), indicating the uniform distribution of L-2-HG.

Related results and discussion were described in the “Results” section as follows:

“Next, LHGFR_{0N3C}, LHGFR_{0N7C}, and LHGFR_{3N7C} were expressed in the cytosol of HEK293FT cells. Addition of 10 mM L-2-HG affected mTFP and Venus fluorescence intensities, which caused non-uniform increase in the emission ratio of LHGFR_{0N3C}-expressing single cell (**Fig. 6a and Supplementary Movie**). The average emission ratio reached a maximum value at 5 min and remained constant during subsequent confocal imaging period (**Fig. 6b**). The apparent K_d of LHGFR_{0N3C} expressed in HEK293FT cells for L-2-HG was determined to be $43.79 \pm 3.05 \mu\text{M}$ (**Fig. 6c**). Based on the emission ratio of non-permeabilized HEK293FT cells under physiological conditions, the basal L-2-HG concentration in LHGFR_{0N3C}-expressing cells was $22.95 \pm 11.22 \mu\text{M}$ (**Fig. 6c**). Only exogenous L-2-HG could significantly increase the emission ratio of LHGFR_{0N3C} in 10 μM digitonin-permeabilized HEK293FT cells, suggesting the specificity of the biosensor inside living human cells (**Fig. 6d**). L2HGDH, the only reported enzyme that is able to catabolize L-2-HG in human cells, is localized in mitochondria¹³. The mitochondrial targeting sequence was appended to LHGFR_{0N3C} to localize the biosensor in mitochondria (**Supplementary Fig. 15a**). Exogenous L-2-HG also induced increase of emission ratio of mitochondrial LHGFR_{0N3C} (**Supplementary Fig. 15b**). Emission ratio of LHGFR_{0N3C} localized in mitochondria was similar to that of in cytosol (**Supplementary Fig. 15c**). The uniform distribution of L-2-HG confirmed the presence of a transporter responding for the transport of L-2-HG between cytosol and mitochondria.”

Figure 6 Monitoring L-2-HG fluctuations in human cells by LHGFR_{0N3C}. **(a)** Sequential images of mTFP (top), Venus (middle), and Venus/mTFP emission ratio (bottom, pseudocolored) of single HEK293FT cell expressing LHGFR_{0N3C}. 10 mM L-2-HG was added at time point zero (min). Elapsed time (in minutes) after addition of L-2-HG is shown at the top of the images. Scale bar, 10 μm . **(b)** Time course of the emission ratio changes inside region of interest (ROI) depicted from the top-left image of **(a)**. **(c)** Normalized dose-response curve of LHGFR_{0N3C} expressed in HEK293FT cells with increasing concentrations (10 nM to 10 mM) of L-2-HG. Cells were permeabilized with 10 μM digitonin. The emission ratio of non-permeabilized HEK293FT cells under physiological conditions is indicated with black dash line. **(d)** Responses of LHGFR_{0N3C} expressed in HEK293FT cells to exogenously added 1 mM glutarate, L-2-HG, D-2-HG, and glucose. Cells were permeabilized with 10 μM digitonin. All data were normalized to the control (ratio in the absence of any tested compounds).

Supplementary Figure 15 Mitochondrial localized LHGFR. **(a)** Confocal microscopy images of mitochondrial LHGFR_{0N3C} (top), LHGFR_{0N7C} (middle), and LHGFR_{3N7C} (bottom)-expressing HEK293FT cells. The images are represented as mTFP channel, Venus channel, and overlay channel from left to right. Scale bar, 10 μ m. **(b)** Emission ratio changes of mitochondrial LHGFR_{0N3C}, LHGFR_{0N7C}, and LHGFR_{3N7C} expressed in HEK293FT cells in response to 10 nM L-2-HG and 10 mM L-2-HG. Mitochondrial LHGFR-expressing HEK293FT cells were permeabilized with 10 μ M digitonin, then 10 nM and 10 mM L-2-HG were added into the treated cell suspension, respectively, and the emission ratios were recorded. Mitochondrial LHGFR_{0N3C} and LHGFR_{0N7C} could respond to L-2-HG application, while LHGFR_{3N7C} could not. The emission ratios were normalized to samples in the presence of 10 nM L-2-HG. $P = 0.0022$, 0.0006, 0.7232 (from left to right). **(c)** Comparison of L-2-HG concentrations between cytosol and mitochondria. The emission ratios of LHGFR localized in cytosol and mitochondria were recorded, respectively, and there was no detectable difference in emission ratio between cytosol and mitochondrial. The emission ratios were normalized to cells expressed cytosolic LHGFR. All data shown are means \pm s.d. ($n = 3$ independent experiments). **, $P < 0.01$ in two-tailed t test; ***, $P < 0.001$ in two-tailed t test; ns, no significant difference ($P \geq 0.05$) in two-tailed t test. Source data are provided as a Source Data file.